# Single-cell sequencing deconvolutes cellular responses to exercise in human skeletal muscle

Alen Lovrić[1,2,3], Ali Rassolie[1,2,3], Seher Alam[1,2], Mirko Mandić[1,2], Amarjit Saini[1,2], Mikael Altun [1,2], Rodrigo Fernandez-Gonzalo[1,2], Thomas Gustafsson[1,2] & Eric Rullman [1,2✉]

Skeletal muscle adaptations to exercise have been associated with a range of health-related benefits, but cell type-specific adaptations within the muscle are incompletely understood. Here we use single-cell sequencing to determine the effects of exercise on cellular composition and cell type-specific processes in human skeletal muscle before and after intense exercise. Fifteen clusters originating from six different cell populations were identified. Most cell populations remained quantitatively stable after exercise, but a large transcriptional response was observed in mesenchymal, endothelial, and myogenic cells, suggesting that these cells are specifically involved in skeletal muscle remodeling. We found three sub-populations of myogenic cells characterized by different maturation stages based on the expression of markers such as *PAX7*, *MYOD1*, *TNNI1*, and *TNNI2*. Exercise accelerated the trajectory of myogenic progenitor cells towards maturation by increasing the transcriptional features of fast- and slow-twitch muscle fibers. The transcriptional regulation of these contractile elements upon differentiation was validated in vitro on primary myoblast cells. The cell type-specific adaptive mechanisms induced by exercise presented here contribute to the understanding of the skeletal muscle adaptations triggered by physical activity and may ultimately have implications for physiological and pathological processes affecting skeletal muscle, such as sarcopenia, cachexia, and glucose homeostasis.

[1] Department of Laboratory Medicine, Section of Clinical Physiology, Karolinska Institutet Huddinge, Huddinge, Sweden. [2] Department of Clinical Physiology, Karolinska University Hospital, Huddinge, Sweden. [3]These authors contributed equally: Alen Lovrić, Ali Rassolie. ✉email: eric.rullman@ki.se

The skeletal muscle is one of the largest organs of the body. In addition to its obvious importance in locomotion, it also plays a central role regulating the body's metabolism. Skeletal muscle is composed mainly of elongated cells called muscle fibers, each of which contains many nuclei. However, other cell types in skeletal muscle have also been shown to be important in maintaining these functions, including cells that infiltrate the muscle from the bloodstream[1–3].

Skeletal muscle is highly plastic, and changes in physical activity lead to a plethora of adaptive processes that, when repeated over time (i.e., training), result in structural and functional adaptations of skeletal muscle[4,5]. Skeletal muscle mass and function have a major impact on physical performance in healthy individuals, but also on the quality of life in old age and on the progression and clinical course of many diseases[6]. The latter is true even when the disease is primarily localized in another organ system, as in heart failure[7,8]. While the molecular machinery underlying the processes of adaptation to exercise has been studied primarily at the level of whole muscle and muscle fibers, information on the effects of exercise on other cell populations in muscle is limited. Existing studies in both animals and humans suggest that many of the cell populations present in muscle are critical for a favorable response to exercise[1,9,10]. For example, immune cells are involved in the recovery response after exercise, fibroblasts appear to have the potential to secrete cytokines involved in various muscle remodeling processes, and endothelial cells produce myogenic and anti-apoptotic factors that promote muscle growth[1].

Single-cell sequencing (scRNA-seq) has been hailed as one of the most important scientific breakthroughs of recent years[11]. With its help, it has been possible to map cellular composition and identify previously unknown cell populations in various tissues. Single-cell studies describing the cellular landscape in different tissues, have yielded some interesting results with potential implications for human physiology. For example, the compendium Tabula Muris (scRNA-seq data) has shown that the niche of mesenchymal cells in mouse skeletal muscle is 3–5 times larger than the niche of satellite cells[12]. Large numbers of mesenchymal cells along with subpopulations of endothelial and satellite cells have also been found in humans, and it has been speculated that a subpopulation of fibro-adipogenic progenitor cells plays a central role in both metabolic adaptations associated with exercise and in aged or diseased skeletal muscle by contributing to increased fibrosis and fat cell infiltration[13,14]. A particularly important cell in skeletal muscle tissue is the myogenic stem cell, also known as satellite cell[1,15]. The satellite cells pool is reduced in aged muscle and in various diseases such as heart failure and diabetes and is accompanied by decreased activation into myoblasts (myogenic progenitor cells) and subsequent proliferation and differentiation capacity[16]. Given the important role of different cell populations within skeletal muscle, a better understanding of cell composition and how different cells in skeletal muscle are affected by different stimuli could provide important insights to fully understand the remodeling processes of peripheral tissues in general and skeletal muscle in particular. This knowledge could also add important information to decipher the pathophysiological mechanisms involved in various diseases related to changes in skeletal muscle mass or function.

In the present study, we use scRNA-seq to characterize single cells from skeletal muscle biopsies obtained before and after a robust remodeling stimulus, i.e., strenuous exercise. The main objective was to characterize the cellular composition of adult skeletal muscle at baseline and to deconvolute the transcriptional responses to exercise in different cell populations, with particular emphasis on myogenic cell subpopulations. We show that scRNA-seq of skeletal muscle quantifies and delineates cell populations in a highly reproducible manner, both in terms of repeatability and across different subjects. Cellular composition of six different main cell types was identified at baseline, with the highest proportion consisted of endothelial cells (44%) followed by mesenchymal cells (26%). Several subpopulations of cells, some not previously described, were also identified, including three distinct populations of myogenic cells and several subpopulations of endothelial, mesenchymal, and immune cells. Most cell populations remained quantitatively stable after three maximal sprint bouts. Immune cells, however, increased substantially, presumably due to infiltration from the circulation. The strongest transcriptional response was observed in mesenchymal, endothelial, and myogenic cells, suggesting that these cells are specifically involved in skeletal muscle remodeling. In agreement with animal studies, we also show that myogenic cells in human skeletal muscle can be divided into three groups characterized by different degrees of cell maturation, and that a single bout of exercise drives these cells towards maturation.

## Results

**Clinical characterization.** Three healthy, moderately active individuals (Table 1) performed three 30 s all-out sprint exercises with 2 min of recovery between sprints. Peak power output in the three sprints was on average 8.87 ± 2 W/kg, corresponding to 10.3 metabolic equivalents (METs). Total energy expenditure was 14.5 ± 6.9 KJ.

**Cellular yield, composition, and biological processes in different cell populations.** The tissue retrieval was 500 ± 150 mg per sample and a total of 39,015 cells from pre- and 24,332 cells from post-exercise were analyzed with a mean read count of 19,507,451 and 19,328 882, respectively. To investigate reproducibility and increase sequence-depth, cells isolated from the third subject were sequenced twice.

**Cell-type annotation and reproducibility.** The six samples from the three subjects, in addition to the two re-sequenced samples from the third subject, were integrated, anchored, and clustered (Fig. 1a). Inter-sample reproducibility with regards to cell-type composition was assessed by a correlation matrix using inter-sample anchoring scores[17]. Agreement both between and within subject was consistent, with anchoring scores ranging from 0.6 to 0.7 (Fig. 1b). All subjects contained cells of all cell types detected (Fig. 1c), with comparable proportions between samples (Fig. 1d). Thus, the cell-type annotations and anchoring were highly reproducible across different subjects and biopsies. Sample distribution across clusters is found in Supplementary Figure 1. Following Leiden-based clustering and cell-type analysis, six major cell types were identified: myogenic cells, endothelial cells, pericyte cells, mesenchymal cells, lymphoid cells, and monocyte cells. Cell-type annotation was based on a combination of marker gene expression compared to the CellMatch database, the scCatch pipeline[18], and an enrichment test of cell types by using highly expressed/marker genes in each cluster relative to the

**Table 1 Study participants.**

| Subject | Age (years) | Weight (kg) | Height (cm) | Sex |
|---------|-------------|-------------|-------------|-----|
| 1st | 26 | 61 | 174 | F |
| 2nd | 50 | 98 | 191 | M |
| 3rd | 26 | 65 | 173 | M |

Three individuals participated in the study. Biopsies from the vastus lateralis muscle were taken before and after a bout of high-intensity exercise.

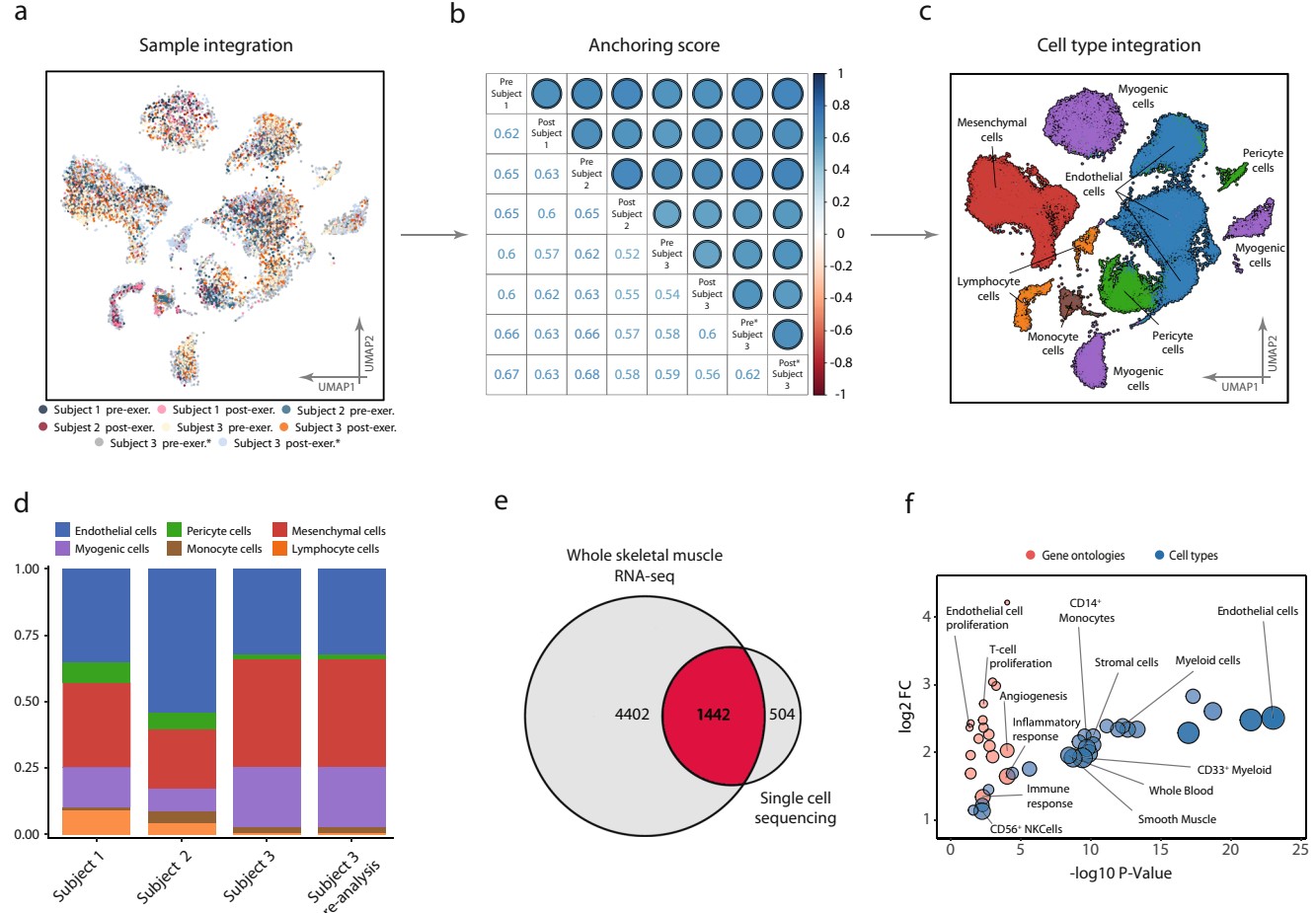

**Fig. 1 Data alignment and reproducibility.** Reproducibility of scRNA-seq between and within subjects. **a** A total of 63 000 cells isolated from six vastus lateralis muscle biopsies from three different subjects were processed, clustered, and visualized using UMAP projection, with color coding indicating the contributing donor and sample. Cells aligned into six major cell populations in which cells from all donors were evenly distributed, providing a measure of reproducibility of cellular composition. **b** This was further confirmed by consistent anchoring scores, a measure of correlation between samples, both within and between subjects, as shown in the correlation-matrix. **c** Cell-type annotation identified six different cell types/lineages as indicated on the UMAP. **d** Cell-type composition in all subjects at baseline, including resequencing of one subject to confirm reproducibility within and between samples. Endothelial cells were the most abundant cell type constituting 44% of the samples followed by mesenchymal 26% and myogenic cells at 18%. Complete composition can be found in Supplementary Data 1. **e** Based on the sequencing data from homogenized whole tissue, 5844 transcripts were identified as part of the skeletal muscle transcriptome and 1946 unique transcripts were detected with the scRNA-seq. A substantial proportion (504 or 26%) of these transcripts were detected only with the scRNA-seq. **f** The unique transcripts from scRNA-seq analysis were highly enriched in genes from endothelial, immune, and stem cell populations, suggesting that scRNA-seq captures a transcriptional signature of cell populations that otherwise goes largely undetected when using RNA-seq strategies.

transcriptomic profiles of cell types provided by the Human Gene Atlas[19,20]. All populations, except the pericyte cluster, were consistently annotated by ≥2 of the methods and were therefore considered unambiguously annotated with respect to cellular origin. Composition tables for individual samples can be found in Supplementary Data 1.

**Comparison with whole-tissue RNA-sequencing.** An obvious advantage of scRNA-seq over whole-tissue RNA-sequencing (RNA-seq) is the ability to deconvolute gene expression profiles into their respective cell-type-based compartments. In addition, it is also possible to provide better coverage of gene expression of less abundant cell types, such as immune and stem cells. To address and objectify this notion, genes detected in the present single-cell experiment were compared with the comprehensive profile of the skeletal muscle transcriptome available through the Genotype-Tissue Expression data (GTEx)[21]. In the 564 skeletal muscle samples from the GTEx study, 5844 unique transcripts

were detected readably, while in the current single-cell experiment a total of 1946 unique transcripts were detected. Of the transcripts detected by scRNA-seq, 504 transcripts were not readably detected in the GTEx data, representing 26% of all the transcripts identified by scRNA-seq (Fig. 1e). Among transcripts detected exclusively in the single-cell experiment, gene and cell ontology analysis revealed strong enrichment of genes derived from immune, endothelial, mesenchymal, and smooth muscle cells (Fig. 1f). Among the differentially enriched biological processes associated with these cell populations were "positive regulation of T cell proliferation" (*fdr* < 0.01), "positive regulation of angiogenesis" (*fdr* < 0.01), and "positive regulation of endothelial cell proliferation" (*fdr* < 0.05), indicative of cell type-specific enrichments (Fig. 1f).

**Cellular composition and cell-type characteristics.** Endothelial cells accounted for 44% of the total number of cells and were distributed in four distinct but neighboring clusters (*ESAM+*,

VWF+/EPS8+, ACKR1+, and FABP4+/NEAT1−) (Fig. 2a). Overall, endothelial cells were characterized by high expression of known endothelial marker genes, including VWF and ESAM (Fig. 2c, d). In comparison with the other cell types, endothelial cells had a significantly higher expression of 165 genes and enriched for 263 gene ontologies such as "regulation vasculature development" (fdr < 0.05), "actin filament organization" (fdr < 0.01), and "response to wounding" (fdr < 0.001) (Fig. 2b). More detailed tables of cell-type and subpopulation specific differential expression together with ontology enrichment can be found in the Supplemental Information.

Mesenchymal cells represented 26% of the total number of cells and showed significantly increased expression of DCN, CFD, and GSN. These cells were further divided into three distinct subpopulations (i.e., DCN+/CFD+, GSN+/LUM+, S100A8+/S100A9+) (Fig. 2a, c, d). The ontologies enriched in the mesenchymal cell population were generic in nature, including "autophagy" (fdr < 0.01), "regulation of anatomical structure morphogenesis" (fdr < 0.01), "oxidative phosphorylation" (fdr < 0.01), "electron transport chain" (fdr < 0.01), and "cell activation" (fdr < 0.05) (Fig. 2b). The S100A8+/S100A9+ subpopulation was considered ambiguously annotated. However, due to the high expression of DCN, VIM, COL6A3, and GSN marker genes the cluster was deemed to be of mesenchymal origin.

Myogenic cells represented 18% of total cells and were further divided into three distinct subpopulations (i.e., PAX7+, TNNI1+, TNNI2+) (Fig. 2a). In addition to PAX7, the first cluster expressed MYF5 and in a smaller frequency MYOD1, indicating this cluster is a mix of undifferentiated satellite cells and early myoblasts. The remaining myogenic subpopulations expressed genes associated with more mature characteristics such as DES, MYL2, and MYOG (Fig. 2c, d), and perhaps terminally differentiated muscle fibers including ACTA1 and MYH6. A subset of these cells also expressed MYOD1. These two subpopulations were distinguished from each other by the expression of slow-twitch (TNNI1) and fast-twitch (TNNI2) troponins. At the ontology level, the myogenic subpopulations exhibited a gene expression profile dominated by key skeletal muscle functions such as "oxidative phosphorylation" (fdr < 0.001) and "mitochondrial respiratory chain" (fdr < 0.001) (Fig. 2b). Consistent with muscle-specific gene expression, the TNNI1+ and TNNI2+ cells also showed enrichment for programs involved in skeletal muscle protein synthesis and degradation (e.g., TRIM63, SYNPO2).

Pericytes accounted for 6% of the total cells, with two distinct clusters (TAGLN+ and NDUFA4L2+) (Fig. 2a). The TAGLN+ cluster was characterized by the high expression of smooth muscle markers such as ACTA2 and pericyte markers (e.g., PDGFR). In addition, this cluster was also annotated as pericyte in origin by scCatch, and therefore pericyte was considered the most probable cellular origin. Significantly enriched ontologies included "mitochondrial electron transport cytochrome c to oxygen" (fdr < 0.001), "nucleoside triphosphate metabolic process" (fdr < 0.001), and "oxidative phosphorylation" (fdr < 0.001) (Fig. 2b).

Two lymphocyte clusters (NAMPT+ and HCST+) were also identified, which accounted for 4% of the total number of cells (Fig. 2a). Considering these clusters presented gene expression profiles characterizing T-, B-, and NK-cells, they were collectively considered as lymphocytes. Genes that distinguished these clusters from the other cell types included CCL5 which is considered as lymphocyte-specific markers (Fig. 2c, d). Ontological enrichment showed "T-cell immunity" (fdr < 0.01), "cell killing" (fdr < 0.05), and "adaptive immune response" (fdr < 0.05).

One monocyte cluster, which accounted for 2% of the total number of cells, was classified as myeloid in origin, based on CD14 expression (Fig. 2a). The cluster differentially expressed genes including CXCL8, AIF, and TYROBP (Fig. 2c, d). HLA-DRA was also highly expressed in this cluster, which is a gene associated with antigen-presenting cells, due to its structural involvement in the formation of human leukocyte antigen (HLA) class II proteins. These CD14+ monocytes enriched for ontologies including "multi organism metabolic process" (fdr < 0.001), "response to corticosterone" (fdr < 0.05), "cellular response to calcium ion" (fdr < 0.05), "response to mineralocorticoid" (fdr < 0.05), "translation initiation" (fdr < 0.001), "Ribosome assembly" (fdr < 0.001), and "Nuclear transcribed mRNA catabolic process nonsense-mediated decay" (fdr < 0.001) (Fig. 2b). Gene-ontology enrichment and marker gene expression for all cell-types can be found in Supplementary Data 2, 3 and 4 and in Supplementary Fig. 2.

**Effects of exercise on cellular composition.** The effects of high intensity exercise were first examined with respect to cellular composition in skeletal muscle (Fig. 3a). Three hours after exercise, circulating cells increased substantially, with lymphocytes increasing from 4 to 9% (p = 0.05) and monocytes from 2 to 4% (p < 0.05), with a corresponding decrease in the relative contribution of resident cells, i.e., endothelial cells decreased from 44 to 37% and pericytes decreased from 6 to 5% after exercise. The proportion of myogenic cells (18%) remained unchanged after exercise.

**Transcriptional response to exercise.** The transcriptional response to exercise was assessed by comparing pre- vs. post-exercise for each cell type separately. A total of 874 (535 unique) genes were differentially expressed (fdr < 0.05) across all different cell types. In terms of number of differentially expressed genes, the mesenchymal cells showed the greatest exercise-related response, with 304 genes differentially expressed (Fig. 3b). In ontological terms, the mesenchymal cells enriched for biological functions involved in tissue regeneration and remodeling, such as "regeneration" (fdr < 0.01), "organ regeneration" (fdr < 0.05), and "wound healing" (fdr < 0.05) (Fig. 3c). Genes driving the enrichment for regeneration biological function in the mesenchymal cells included VIM, UBC, GPX4, and AVCRL1 (Fig. 3d). Several genes involved in cytoskeletal reorganization and cell-cycle activation, such as RHOBTB3, TPM1, and RGCC, were also robustly upregulated after exercise in this cell type.

The endothelial cells showed the second greatest response to exercise with a total of 281 differentially expressed genes (Fig. 3b). The main ontological characterization included cell activation and stress reactions, such as "cell cycle G2 M-phase transition" (fdr < 0.05), "energy reserve metabolic response" (fdr < 0.01), and "tissue regeneration" (fdr < 0.05) (Fig. 3c). Differentially expressed genes such as TIMP3, ACTB, UBC, and CALM1 (Fig. 3e) indicate endothelial re-composition and stress response after exercise.

The myogenic cell populations differentially expressed 111 genes (fdr < 0.05) after exercise, including genes involved in differentiation along myogenic lineage such as MYOD1, and MYF6 (Fig. 4). In the undifferentiated PAX7+ cluster gene ontologies related primarily to cellular stress response, such as "negative regulation of cell death" (fdr < 0.001) and "regulation of growth" (fdr < 0.05) (Fig. 3c). The genes driving the stress-related enrichments for the PAX7+ cluster included UBC, HSP90AB1, LMNA, NCL, and SOD2. Muscle-related functions, such as "actin-mediated cell contraction" (fdr < 0.001), and "muscle system process" (fdr < 0.001), were enriched in the more mature clusters (TNNI1+, and TNNI2+) (Fig. 3c) and mainly driven by gene expression such as DES, ACTA, MYL2, and MYOZ1.

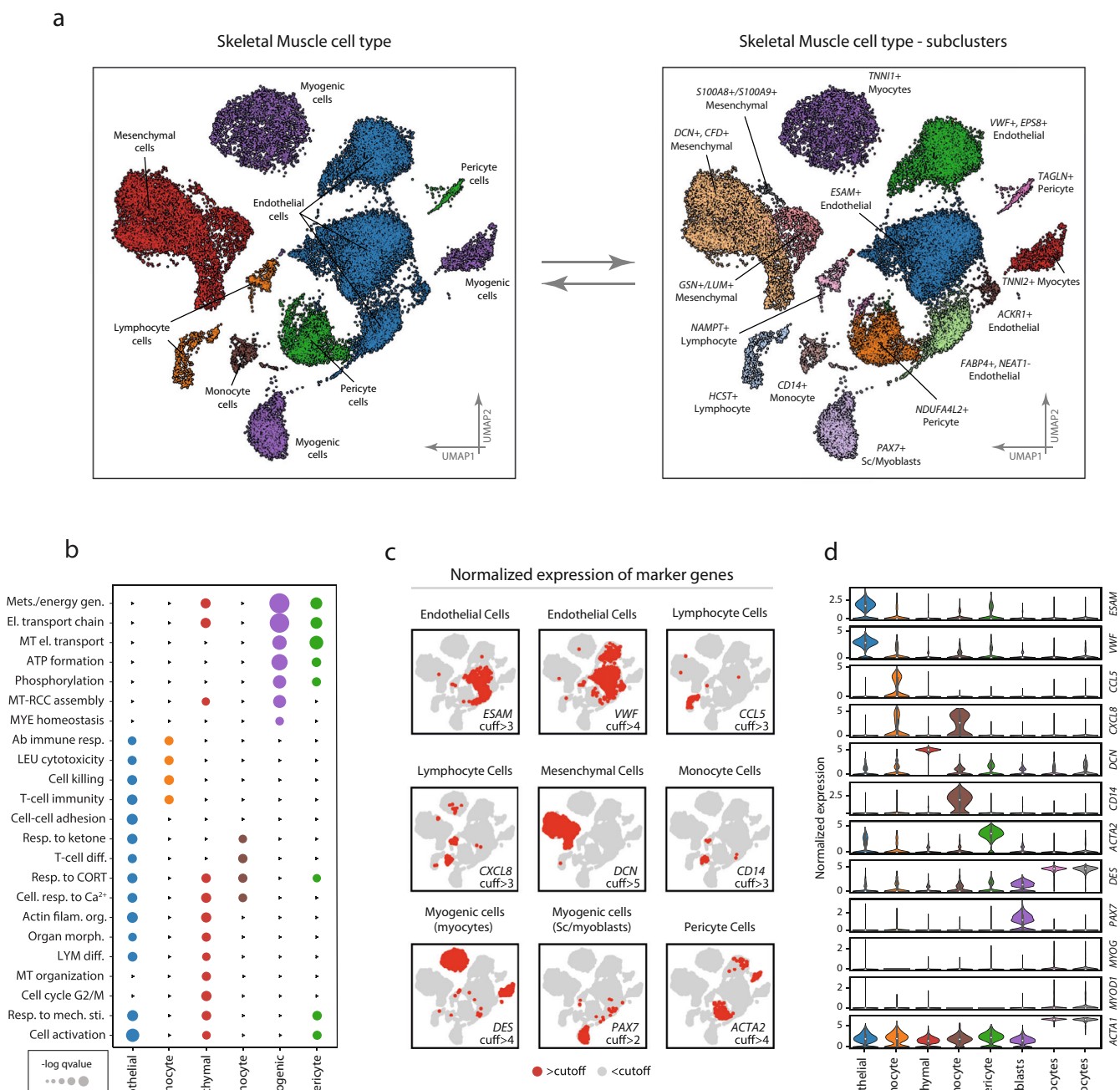

**Fig. 2 Cell-type and subpopulations.** Marker expression, and ontology enrichment across the different cell types and subpopulations. **a** Cell-type annotation was based on a combination of marker gene expression compared with the CellMatch database in relation to profiles of specific cell types in the Human Gene Atlas. Six different cell types/lineages were identified and color-coded on the UMAP: Endothelial cells, Myogenic cells, Mesenchymal cells, Myeloid cells, Lymphoid cells, and Pericyte cells. Distinct cell clusters derived from the same cell type were compared and characterized through differential expression. Four subpopulations of endothelial cells were identified to differ by the expression of *VWF+/EPS8+*, *ESAM+*, and *FABP4+/NEAT−*, respectively. There were three mesenchymal subpopulations characterized by differential expression of *DCN+/CFD+*, *GSN+/LUM+*, and *S100A8+/S100A9+*, and muscle cells could be further subdivided into three myogenic subpopulations, including *PAX7+* satellite/myoblast cells, and fast- (*TNNI2+*) and slow-twitch (*TNNI1+*) troponin expressing cells. Pericytes characterized by high/marker gene expression of *ACTA2* and *PDGFR* were divided into two subpopulations (*NDUFA4L2+* and *TAGLN+*). One monocyte population was found to express *CD14*. **b** Enrichment analysis of biological functions in the different cell populations shows functionally different gene expression profiles for the different cell types. Color coding represents different cell-type and larger dot radius denotes more significant biological function (*fdr* < 0.05). **c** Marker gene expression in the different cell populations. Red and gray colors indicate the expression of each marker gene above and below hard threshold in all cells, respectively. **d** Quantitative comparison of the expression of the marker genes in the different cell types. The data in boxplots nested within violin plots are expressed as median, interquartile range, minimum and maximum values. Normalized expression refers to log(1 + x) if not stated otherwise. SC satellite cells; ST slow-twitch, FT fast-twitch, cuff cutoff. Complete marker-gene differential expression can be found in Supplementary Data 2–4.

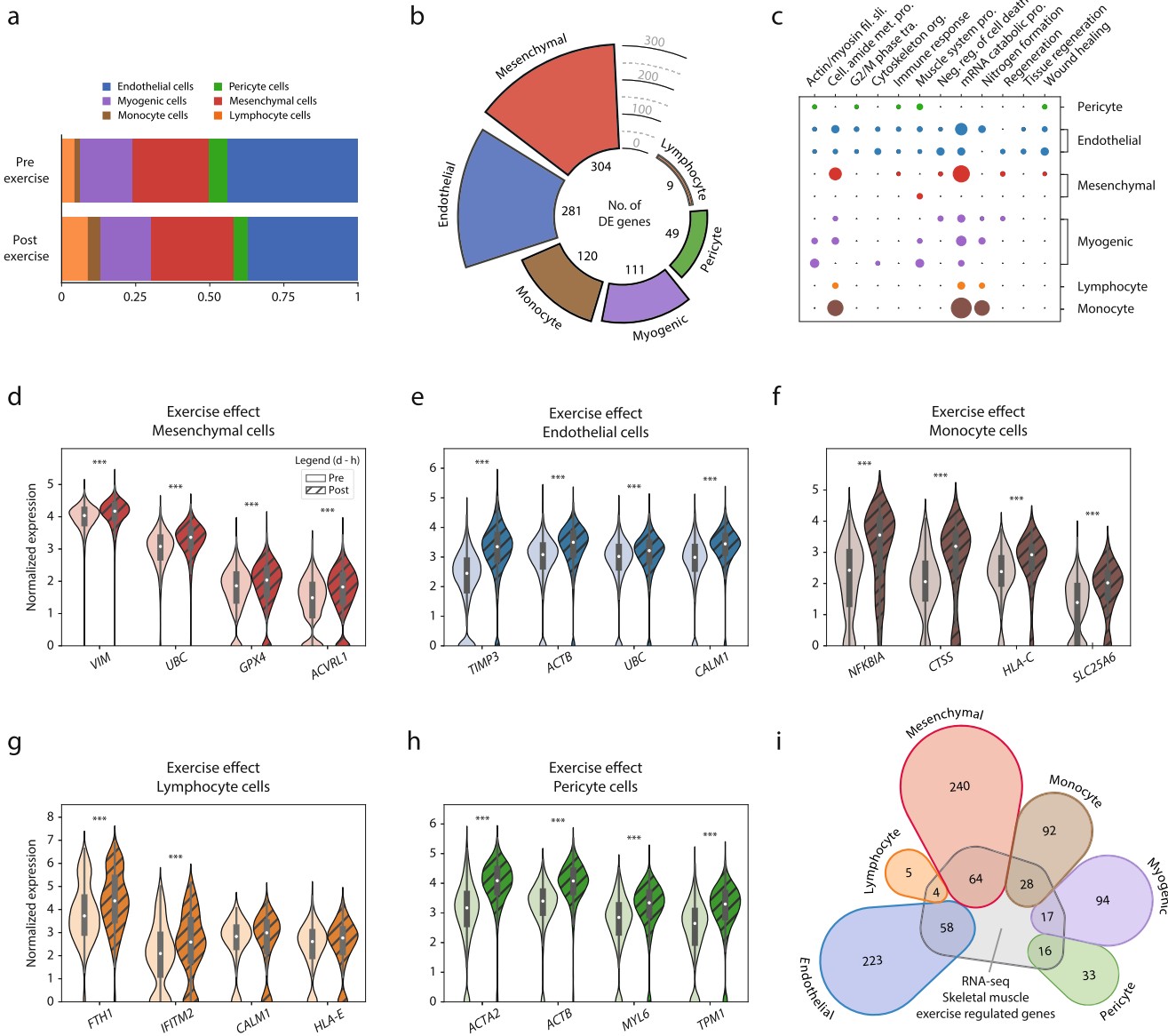

**Fig. 3 Transcriptional response to a single bout of exercise across cell types.** Exercise effects detected by scRNA-seq analysis of the human skeletal muscle. **a** Cellular composition analysis of samples before (pre-exercise) and three hours after (post-exercise) a single bout of exercise. **b** Differential expression analysis across cell types after a single bout of exercise. Color coding represents cell-type, and bar height denotes the number of exercise-regulated genes. In total, 874 (535 unique) genes were upregulated by exercise, with mesenchymal cells having the highest number (304), followed by endothelial cells (281) and monocytes (120). In contrast, only 9 genes were found to respond to the exercise stimulus in the lymphocyte population. **c** Gene-ontology analysis revealed that most biological processes regulated by exercise were cell-type specific, with a small number of processes similarly regulated in most cell types ("mRNA catabolic process" and "nitrogen formation"). Color coding represents different cell-type, and a larger dot radius denotes a more significant gene ontology ($fdr < 0.05$). **d**–**h** Normalized expression level of representative genes for each cell type that were differentially expressed after exercise. The data in boxplots nested within violin plots are expressed as median, interquartile range, minimum and maximum values. \*\*\*$fdr < 10e-5$. **i** Genes that were significantly regulated by exercise in the current scRNA-seq experiment were compared with differentially expressed genes after exercise using RNA-seq from whole muscle. Color coding represents different cell-type, and numerical value denotes the number of differentially expressed genes shared by both methods or those exclusively detected by scRNA-seq. Approximately 25% of the genes detected by scRNA-seq were also identified using RNA-seq and this was consistent across all cell types. Normalized expression refers to $\log(1 + x)$ if not stated otherwise. Complete exercise-differential expression and ontology analysis can be found in Supplementary Data 5–9.

The monocyte cell population differentially expressed 120 genes after exercise (Fig. 3b). Ontologically, the monocytes presented a generic response to exercise, with terms such as "amide biosynthetic process" ($fdr < 0.001$), "protein targeting" ($fdr < 0.001$), and "peptide metabolic process" ($fdr < 0.001$) (Fig. 3c). When considering the differential expression of single genes, the monocyte population significantly expressed *NFKBIA*, *CTSS*, *HLA-C*, and *SLC25A6* after exercise, suggesting

an inflammatory response and cellular stress reactions to exercise (Fig. 3f). Lymphocytes differentially expressed 9 genes after exercise (Fig. 3b). Ontologically, the lymphocyte cells solely enriched for generic terms, suggesting an absent exercise-specific response. The genes regulating the ontological enrichment were overall generic and lacked established biological functions in the literature (e.g., *FTH1*, *IFITM2*, *CALM1*, and *HLA-E*) (Fig. 3g).

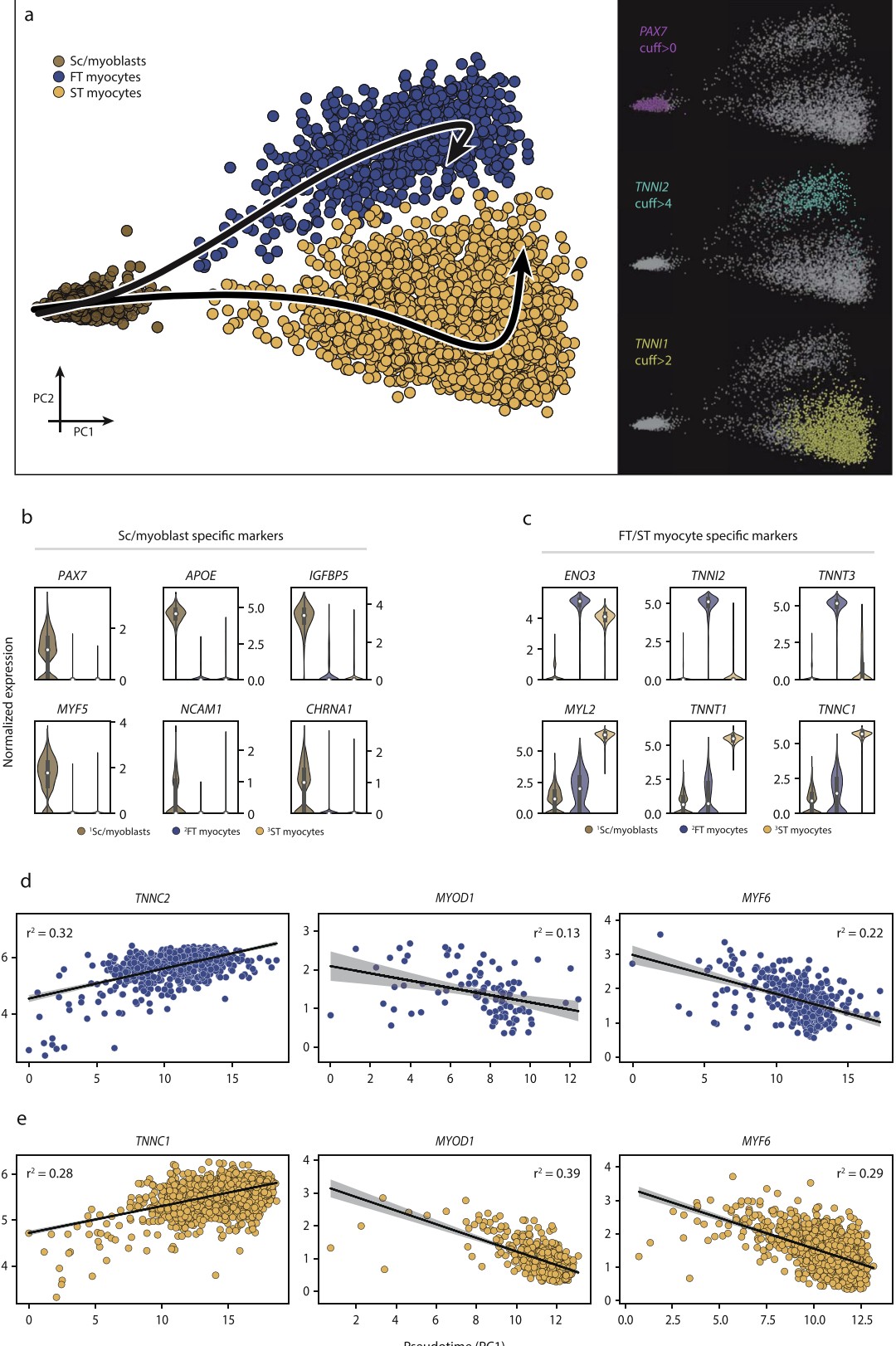

Pericytes differentially expressed 49 genes (Fig. 3b). In terms of ontological enrichment, the pericytes showed a response indicative of cellular activation, stress response, and regeneration, enriching for 38 ontologies such as "response to wounding" (*fdr* < 0.01), "actin mediated cell contraction" (*fdr* < 0.05), and "cell cycle G2 M phase transition" (*fdr* < 0.05) (Fig. 3c). Genes driving the regenerative response included *ACTB*, *TPM1*, *TIMP3*, and *CD36*. In terms of the greatest exercise-related response, the pericytes differentially expressed *ACTB* and *ACTA2* after exercise (Fig. 3h). A complete list of differentially expressed genes and ontologies across cell-populations can be found in Supplementary Data 5–8.

**Fig. 4 Myogenic cell trajectories.** Three distinct myogenic subpopulations of myogenic cells were identified and further investigated for their distinguishing features and trajectories. **a** The first cluster (brown) was characterized by *PAX7*+ expression (satellite/myoblast cells), whereas the remaining two myogenic subpopulations expressed higher levels of genes indicative of maturation, including slow-twitch (*TNNI1*) and fast-twitch (*TNNI2*) troponins, along their respective trajectories. **b**, **c** Highly expressed genes in the undifferentiated *PAX7*+ stem/progenitor cell subpopulation relative to the more differentiated subpopulations. The undifferentiated cell markers included *PAX7*, *NCAM1*, *MYF5*, and *APOE*, while more differentiated cells expressed *ENO3*, *TNNI1*, *TNNI2*, and *MYL2*. The data in boxplots nested within violin plots are expressed as median, interquartile range, minimum and maximum values. **d**, **e** There was a successive increase in *TNNC1* and *TNNC2* expression as cells adopted higher absolute values along the trajectory, indicating a continuous differentiation process within each subpopulation. In parallel, the expression of several other genes involved in differentiation (*MYOD1*, *MYF5*) decreased, consistent with the current understanding of the cell maturation along the myogenic lineage. Shaded area indicates estimated 95% confidence intervals for the regression estimate. Normalized expression refers to log(1 + x) if not stated otherwise. SC satellite cells, ST slow-twitch, FT fast-twitch, cuff cutoff.

**Comparison of single-cell with whole-tissue sequencing in relation to exercise**. We also examined whether the transcriptional responses to exercise among different cell populations using scRNA-seq were consistent with RNA-seq findings. To this end, the transcriptional responses to exercise in each cell type were compared with findings from a recent RNA-seq study[22] investigating the same time-points and with a similar exercise protocol as in the current study. Of the 874 transcripts (535 unique) that were significantly regulated in ≥1 cell-type in the current scRNA-seq experiment, 187 transcripts (129 unique) were also regulated by exercise in RNA-seq study (Fig. 3i). There were no differences between cell populations in terms of coverage by RNA-seq, i.e., all cell types shared ~25% of transcripts regulated by exercise with RNA-seq.

**Trajectory analysis of myogenic cells**. A major advantage of scRNA-seq is the ability to use the global transcriptome of each cell to classify populations of cells of common origin into a continuum of only 1 to 2 dimensions, thereby visualizing and identifying successive, stepwise changes in gene expression from cell to cell. This technique is often referred to as trajectory analysis. Trajectory analysis has proven to be a powerful tool to deconvolute successive transcription-driven cellular processes in developmental biology and stem cell differentiation. Here, we use trajectory analysis to test whether there is evidence of a continuous transition from undifferentiated myogenic cells into increasingly mature myogenic cells (Fig. 4a). Three distinct clusters corresponding to undifferentiated *PAX7*+ satellite/myoblast cells, *TNNI2*+ fast-twitch, and *TNNI1*+ slow-twitch myogenic cells were selected, and two different trajectories with the undifferentiated cluster as a starting point were calculated using principal curve pseudotime analysis. Genetic markers that determined position along the first principal component included *PAX7*, *APOE*, *IGFBP5*, *MYF5*, and *NCAM1* (Fig. 4b), which are canonical myogenic stem/progenitor cell markers. Among the more differentiated cells, *ENO3*, *TNNI2*, *TNNT3*, *MYL2*, *TNNT1*, and *TNNC1* genes (Fig. 4c) drove the partitioning into distinct clusters along the second principal component. The transition of undifferentiated *PAX7*+ satellite/myoblast cells towards a higher degree of differentiation was proportional to expression levels of marker genes along the given trajectories. Accordingly, the expression of *TNNC1* and *TNNC2* increased proportionally to pseudotime, in slow- and fast-twitch myogenic cells, respectively ($r^2 = 0.28$ and $r^2 = 0.32$, Fig. 4d, e). A corresponding decrease in the expression of *MYOD1* ($r^2 = -0.39$ and $r^2 = 0.13$) and *MYF6* ($r^2 = -0.29$ and $r^2 = -0.22$) was observed along the trajectories for both fast- and slow-twitch cells (Fig. 4d, e).

Furthermore, we examined the effect of exercise on the transition from *PAX7*+ undifferentiated satellite/myoblast cells to fast- and slow-twitch expressing cells and the overall transcriptional effect of exercise in these subpopulations. Three hours after a single bout of exercise, there was a small ($\Delta_{slow-twitch} = 5.4\%$, $p < 0.001$; $\Delta_{fast-twitch} = 9.0\%$, $p < 0.001$) but statistically significant

incremental shift in pseudotime toward a higher degree of differentiation in both the slow- and fast-twitch cell populations (Fig. 5a, b). Apart from the effects on pseudotime, there was a common transcriptional response to a single bout of exercise for 24 genes in all three myogenic subpopulations. These genes included mostly mitochondrial and ribosomal genes. More genes were regulated by exercise in the undifferentiated *PAX7*+ (255 genes) and slow-twitch (138 genes) subpopulations, compared to the fast-twitch subpopulation (51 genes) (Fig. 5c). Genes regulated by exercise in the undifferentiated *PAX7*+ myogenic cells included *NEAT1*, *NNMT*, *CXXC5*, *MT2A*, and *SQSTM1* (Fig. 5d). The slow-twitch *TNNI1*+ cells responded to exercise by regulating genes involved in "muscle organ development" (*fdr <* 0.001) including *MYLPF*, *ACTA1*, *MYL2*, and *MYH7* (Fig. 5d). In the fast-twitch *TNNI2*+ cells, the exercise response included upregulation of changes in *MYBPC1*, *MYBCP2* ("muscle contraction", *fdr <* 0.001), *TXNIP*, *UBC*, and *OPTN* (Fig. 5d). Gene-ontology enrichment in the myogenic cells following exercise can be found in Supplementary Data 9.

**In vitro validation of myogenic subpopulations**. The observation of a regulated expression of contractile elements such as *TNNC1* and *TNNC2* in myogenic cells, and that such transcriptional upregulation is associated with the initiation of differentiation towards a more mature myofiber-like phenotype was validated in vitro. Primary myoblasts were isolated from human muscle biopsy and kept in proliferation media until confluence. Differentiation towards myotube formation was initiated according to standard protocols and the expression of key-marker genes from the myoblast subpopulations identified as undergoing differentiation were analyzed with qPCR on day 0, day 4, and day 9 of the differentiation process (Fig. 5e). Slow-twitch troponin (*TNNC1*) increased from $9.3 \pm 2$ a.u on day 0 to $331 \pm 150$ after 4 days of differentiation ($p < 0.001$). It remained elevated after 9 days of differentiation ($117 \pm 20$) ($p < 0.001$). Gene-expression of fast-twitch troponin (*TNNC2*) was $2.2 \pm 1$ a.u on day 0, increased to $123 \pm 10$ a.u after 4 days of differentiation ($p < 0.001$), and remained elevated after 9 days of differentiation ($155 \pm 30$) ($p < 0.001$).

Finally, we utilized a publicly available microarray experiment conducted in a mouse myoblast cell-line (C2C12-cells) evaluation gene-expression in cells undergoing differentiation ($n = 3$) in relation to cells in proliferation-media ($n = 3$) (Fig. 5f). *TNNC1* and *TNNC2* were elevated with a log2FC of 5.7 and 7.7 respectively (*fdr <* 0.001) in differentiating versus proliferating C2C12-cells.

**Discussion**

Here we report the characterization of the cellular composition of adult human skeletal muscle using scRNA-seq and show that it is reproducible within and between different individuals. We also demonstrate changes in cell composition and transcriptional

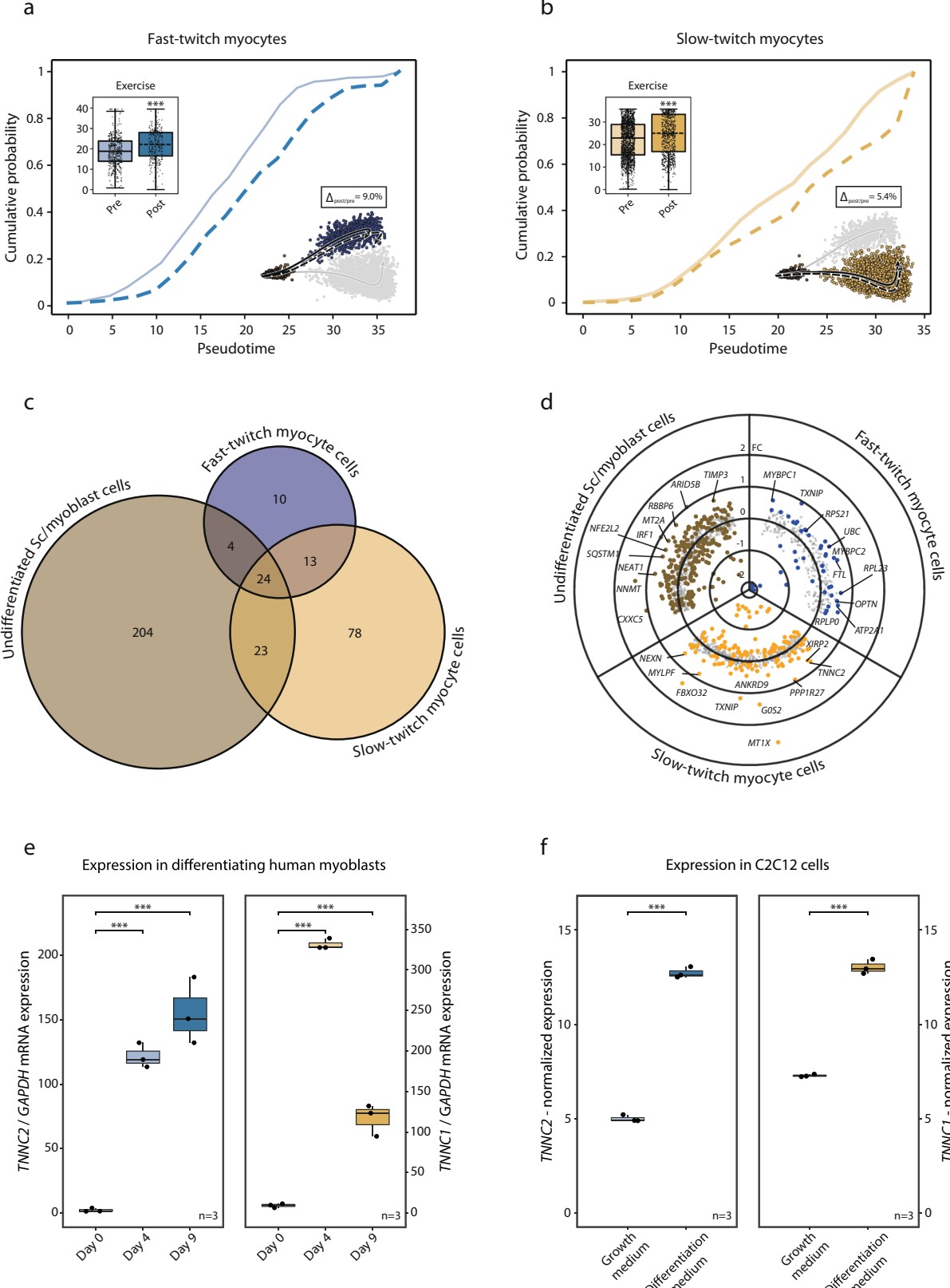

responses to exercise in different cell populations of skeletal muscle. We identify three distinct subpopulations of myogenic cells and use trajectory analysis to show that a continuous process of myogenic differentiation and maturation occurs in resting human skeletal muscle. Finally, we show how a single exercise session accelerates the myogenic trajectory toward a more mature

transcriptional signature consistent with fast- and slow-twitch myofibers.

The single-cell atlas projects have had a tremendous impact on understanding the numbers and properties of cell populations in different tissues. However, due to the global, whole-body nature of these studies, the number of cells characterized, and the

**Fig. 5 Exercise effects on myogenic cells.** The effect of exercise on the myogenic cell populations fate was further investigated: **a, b** Three hours post-exercise there was a significant change along the trajectories with a shift of 9.0 and 5.4% ($p < 0.001$ for both) in fast- and slow-twitch myogenic cells respectively, indicating increased differentiation. Boxplots and ECDFs denote the position along pseudotime of the myogenic cells pre- vs. post-exercise. **c** Venn diagram of exercise-regulated genes in the myogenic subpopulations. The larger exercise effect was observed among the undifferentiated PAX7+ satellite/myoblast subpopulation compared to the more mature subpopulations. A substantial portion of the exercise-regulated genes in the more mature fast- and slow-twitch subpopulations were also regulated in the undifferentiated PAX7+ subpopulation. **d** Scatter/volcano plot presenting the transcriptional effects of exercise for each myogenic subpopulation. Differentially expressed genes (fdr < 0.05) are highlighted where distinct color denotes respective subpopulation. **e** In vitro validation experiments through analysis of gene expression of slow- and fast-twitch troponins in primary human myoblasts undergoing differentiation. Boxplots depict gene-expression of TNNC2 and TNNC1 mRNA levels assessed through RT-PCR at baseline, after 4 and 9 days of differentiation towards myotubes. Statistical analysis was conducted through One-way ANOVA with the Tukey test as a post hoc. **f** C2C12 cells in proliferation versus differentiation media with gene expression analyzed using microarrays obtained through LIMMA. Where present the data in boxplots are expressed as median, interquartile range, minimum and maximum values, and individual points are shown as black dots. Normalized expression refers to log2(x) if not stated otherwise. SC satellite cells.

sequencing depth used for each cell in the different tissues have been rather limited. Hence, there is a need for a deeper and more detailed analysis of specific tissues and in relation to physiology, alongside methodological validation. The current study demonstrated that scRNA-seq analysis of skeletal muscle can quantify and delineate cell populations in a reproducible manner. The cell yield (i.e., the number of viable cells recovered per unit of tissue) varied considerably across samples, but it was within the range of recovery rates seen for other techniques, such as FACS or isolation of cells for culture[23]. Despite this variation, the relative proportion of different cell populations in different samples was consistent both between different subjects and when samples from the same subject were re-sequenced (i.e., repeatability). The fact that all donors contributed comparably to all cell populations supports the validity of previous single-cell characterizations of muscle tissue in which cells from multiple donors were pooled or were based on data from single individuals[13,14]. However, it should be noted that attempts to quantify differences in cell density between different samples using scRNA-seq should be treated with caution due to the high variance in recovery rates.

The baseline cellular composition of skeletal muscle was characterized by endothelial cells, which accounted for the largest proportion (44%), followed by mesenchymal cells (26%), myogenic cells (18%), and pericytes (6%). Undifferentiated myogenic progenitor cells accounted for 6% of the total cells in muscle, which is consistent with the proportion of these cells in adult human skeletal muscle[24]. Immune cells accounted for only a small proportion of the extracted pre-exercise cells in this study, namely lymphocytes (4%) and monocytes (2%), which is also consistent with the previously determined proportions of these cell types in skeletal muscle[25]. Thus, the relative contribution of most cell types reported here is coherent with previous studies that have analyzed tissue morphology using microscopic techniques. Overall, our data indicate that scRNA-seq gives a good representation of the mononucleated cell types present in the muscle.

Exercise is a well-known stimulus to provoke significant structural and functional adaptations in skeletal muscle. Based on this, we selected a very strenuous exercise routine (repeated sprint cycling at maximal intensity) with well-known large local adaptations in the skeletal muscle, to investigate its effects on cell composition and transcriptional response in different cell populations in skeletal muscle. The different cell populations remained quantitatively stable after the exercise, with the exception of an increase in lymphocytes, which confirmed previous reports using microscopy techniques[26]. Despite the substantial increase in the number of cells, the lymphocyte cell population had the lowest number of exercise-regulated genes and therefore we suggest that the main mechanism behind this increment in lymphocyte number after exercise is an effect of post-exercise skeletal muscle

hyperemia, rather than active extravasation. In any case, since the number of resident lymphocytes and monocytes is low, future experiments with active enrichment of immune cells prior to sequencing could increase the resolution and thus identify subpopulations of lymphocytes cells with different or more active response to exercise.

The transcriptional response of skeletal muscle to exercise has been extensively studied by us and other groups[27–29]. These experiments have used RNA-seq and microarray techniques, or qPCR, and have generally assumed that most of the mRNA that is measured originates from mature myofibers. This assumption is supported herein, where 25% of the genes detected with scRNA-seq are not abundantly detected using RNA-seq. These genes were mainly originating from mononuclear cell populations such as lymphocytes, myogenic, and mesenchymal cells. In general, the transcriptional response to sprint exercise varied greatly between cell populations, and very few genes and gene ontologies were regulated in a similar manner in more than 1 or 2 cell populations. The lack of a universal gene expression to sprint exercise in the different cell types may also explain why most of the genes regulated by exercise in both scRNA-seq and RNA-seq were highly enriched for generic biological functions such as 'Transcriptional regulation' and 'Ribosomal biogenesis'. This underscores the idea that there are very few, if any, genes/processes that respond uniformly across cell types in skeletal muscle in response to exercise.

When looking at specific cell population responses to the exercise stimulus employed here, the strongest transcriptional changes were observed in mesenchymal cells, followed by endothelial, monocyte, and myogenic populations. In addition to the transcriptional response to exercise, the mesenchymal cell population also represented a rather large proportion of the skeletal muscle-derived cells, albeit divided into several distinct subpopulations. The latter might be related to the heterogeneity of these cells in terms of classification and functionality[30,31]. However, a common feature of mesenchymal cells is their importance for regeneration in different tissues, in particular their role in remodeling the extracellular matrix[32,33]. Although the importance of the extracellular matrix in the remodeling process is gaining scientific attention, it has been underestimated in the past compared to other mechanisms. The fact that mesenchymal cells showed the strongest transcriptional response to exercise, together with the enrichment of genes with biological functions involved in tissue regeneration and remodeling reported in the current study, seem to support the importance of mesenchymal cells and extracellular matrix in skeletal muscle remodeling. Furthermore, extracellular matrix remodeling provides a physical link between vascular cells and their surrounding tissues, and it is suggested to orchestrate increased capillarization (angiogenesis)[34]. It follows that such angiogenic-induced tissue

remodeling is mainly driven by endothelial cells. Therefore, it was not surprising that sprint exercise elicited a very strong effect on endothelial cells transcription machinery. This transcriptional response varied between subpopulations of endothelial cells, suggesting that the different populations have diverse roles in maintaining the vascular functions of capillaries, including the response to an angiogenic stimuli (i.e., repeated sprint exercise).

The relative increase in the number of monocytes in skeletal muscle tissue after sprint exercise was rather small, but in line with what has been reported in microscopy-based studies. The small increment in numbers was, however, accompanied by a profound transcriptional response in monocytes, which contrasts with the lack of transcriptional changes observed in the lymphocyte population after exercise[25,35]. These changes included a strong upregulation of several adhesion molecules and cell cycle markers, suggesting that the monocyte cell population was qualitatively influenced toward a proliferative stage in the hours following an exercise bout. Overall, these changes imply an active infiltration of monocytes from blood into the muscle, and a subsequent transition regulation of pro-inflammatory mediators such as *NFKB* and *CXCL2*, as well as the translatory machinery driven by ribosome-expression.

The satellite and myoblast cell shares the basal lamina with the skeletal muscle cell and is therefore well positioned to receive signals from the myofibre, as well as from the local milieu. Since its discovery in 1961, great efforts have been made to elucidate its role, the mechanisms of activation, and their final fate[36]. Recent single-cell studies have put forward several subpopulations of myoblasts residing in the muscle of healthy individuals at rest[13,37] and differences in response upon muscle injury[38]. Here we found three distinct myogenic subpopulations of muscle cells with distinct responses to exercise. Traditionally, the key function of myoblasts has been assumed to be fusion with the mature muscle fiber upon muscle injury, but in later years new roles involving the coordination of remodeling through the release of various cytokines have been proposed[1]. Satellite cells are known to be activated by microinjuries or muscle tension-induced signals caused by many types of exercise[15]. Consistent with this, the current protocol of maximal repeated sprint cycling induced robust transcriptional changes in the muscle populations, of which ~300 transcripts were associated with cell cycle, differentiation, and transcriptional regulation.

Trajectory analysis of the muscle cells demonstrated that two of the three myogenic subpopulations had a successive increase in genes associated with differentiation toward slow- or fast-twitch fibers, indicating an ongoing differentiation process. The finding of myogenic subpopulations characterized by differential expression of contractile elements, which we assume is caused by different stages of early differentiation, was validated in vitro on primary myoblasts cells and in a C2C12 cell-line. In line with this concept, in human primary myoblasts, as well as in C2C12 cells, there was a robust (logFC ~5) increase in gene-expression of both fast- and slow-twitch troponin when cells were put in differentiation media. This confirms that the regulation of these genes is part of the early stages of differentiation in myoblasts and while primary cultures and single-cell clusters cannot be assumed entirely pure in terms of cell-type and a contribution of terminally differentiated myonuclei cannot be ruled out, the C2C12 data show that this is indeed a characteristic of the early differentiating myogenic cell. This continuous upregulation of slow- and fast-twitch characteristics was paralleled by a corresponding downregulation of transcripts associated with early stages of satellite cell differentiation, such as *MYOD1* and *MYH8*[39,40]. Following exercise, there was a significant shift along the trajectory indicating a surge in the transcriptional programs associated with maturity in both fast- and slow-twitch (*TNNI2*+ and *TNNI1*+)

subpopulations. Interestingly, this shift was somewhat larger, and more genes were regulated by exercise, in the slow-twitch *TNNI1*+ compared to fast-twitch *TNNI2*+ subpopulation. Given the phenotypic adaptations associated with repeated sprint training favoring a more oxidative muscle phenotype, it is tempting to speculate that the first steps of this adaptive process are already visible in the transcriptome of individual myoblasts after a single exercise session. However, whether the trajectory data reported here influences the skeletal muscle phenotype, i.e., the increase in myonuclear content and modification of slow- and/or fast-twitch fiber composition over time on a stimulus-dependent manner, remains to be confirmed. Overall, the current study supports the use of scRNA-seq to better characterize the cell-specific responses to exercise in human skeletal muscle tissue. Given that this tissue is available for analysis and can be remodeled in a controlled manner, this strategy could be a powerful addition to experiments performed in animals and cells and could provide new information about peripheral remodeling processes, including plausible cell-cell interactions.

The present study is focused on outlining the immediate effects of a single bout of exercise on the cellular transcriptional landscape in the skeletal muscle, but it is important to underscore that the study has several limitations with regards to generalizability due to the small number of subjects investigated. This means that only baseline features and exercise effects that are preserved in all subjects can be delineated and potential differences across subjects' sex and age may pass undetected. Despite this limitation, we have shown that scRNA-seq analysis of skeletal muscle tissue is capable of quantifying and delineating cell populations in a highly reproducible manner, both in terms of repeatability and across different subjects. The scRNA-seq analysis presented in this study provided additional information to RNA-seq analysis in terms of transcriptional response in cell types other than the muscle fibers. We show that a single bout of exercise leads to distinct cell type-specific transcriptional responses where the strongest response was observed in mesenchymal, endothelial, and myogenic cells, suggesting that these cells are specifically involved in skeletal muscle remodeling. Finally, we show that myogenic cells in human skeletal muscle can be divided into three groups characterized by different degrees of cell maturation, and that exercise stimulates subpopulation of undifferentiated stem/progenitor myogenic cells to mature toward slow- or fast-twitch fibers, a process that may be involved in the specific phenotypic adaptations induced by the stimulus used in the current experiments.

## Methods

**Study participants**. Three healthy individuals (one female and two males) were recruited for the study by advertisements at the university and local community. Inclusion criteria were healthy adults 18–50 years of age, physically active on a regular basis (at least twice per week). Exclusion criteria were active medication of any kind, participation in athlete-level organized sports and regular use of any tobacco products. Subjects fulfilling these criteria announcing interest to participate were enrolled on a regular basis until three experiments had been conducted. The study was conducted in alignment with the declaration of Helsinki and was approved by the National Swedish ethical council (accession number 2019-04027), oral and written informed consent was obtained from all participants.

**Exercise bouts and biopsies**. Each subject underwent two muscle biopsies of the vastus lateralis using the Bergstrom technique[41] (one from each leg in randomized order) extracting approximately 1 g of tissue, one biopsy immediately before and one 3 h post-exercise. The exercise intervention comprised 3 × 30 s all-out sprints on a mechanically braked cycle ergometer (Monark 894E, Varberg, Sweden) against a breaking force equivalent to 0.075 kg/kg body weight, as previously described[42]. In brief, preceding the first interval each subject completed a short warm-up (2.5 min), followed by 3 sprint intervals each separated by 2 min of unloaded cycling. The final interval was followed by 2 min of unloaded cycling serving as a cool-down. Immediately after collection, the muscle biopsies were cleaned of visible adipose and connective tissue and were then minced into small pieces (<1 mm3) in basal medium DMEM-F12 GlutaMAX (Gibco Invitrogen) containing 0.5% human serum albumin, 1% ABAM, collagenase B (2 mg/ml, cat

no. 11,088 807,001, Roche, Germany) and dispase II (2 mg/ml, Sigma, cat. no. D4693-1G) and incubated for 1 h in an orbital shaker at 150 rpm and 37 °C with trituration every 15 minutes to dissociate muscle-derived cells. Enzymatic dissociation was terminated by placing the sample on ice for 5 min and the cell suspension passed through 70, 40, 30, and 20 µM pluriStrainers (pluri Select) to remove myofiber debris. Removal of cellular debris from viable cells was performed on the filtered cell solution by density gradient using Debris Removal Solution (Miltenyi Biotech, cat no. 130-109-398) according to manufacturers' instructions and cells were resuspended in 1 ml PBS containing 1% human serum albumin. Removal of red bloods cells from the cell suspension was performed using Red Blood Cell Lysis solution (Miltenyi Biotech, cat no. 130-094-183) following manufacturers' instructions. The solution ensures optimal lysis of erythrocytes with minimal effect on all cell types obtained from tissue samples. Cells were resuspended in 400 µl PBS containing 1% human serum albumin and passed through a 20 µM pluriStrainer into a 1.5 ml Eppendorf tube pre-coated in PBS/10% human serum albumin to prevent cell adherence to the container surface. Samples were stored on ice prior to single-cell RNA sequencing in a 10X core facility for library preparation and high-resolution single cell sequencing based on the well-established 10X protocol.

**Statistics and reproducibility**. The experiments are based on the three subjects with samples from two timepoints and all statistics on organism, tissue, and cell-population is based on these replicates. Within (pre vs post exercise) and between cell-population differential expression analysis considered cells as replicates with ranked-order non-parametric statistics as outlined below. Cell-validation experiments are based on $n = 3$ biological replicates for both human primary and C2C12-cells.

**Mononuclear cell single-cell sequencing**. For each sample, an aliquot of cells was taken and stained for viability with calcein-AM and ethidium-homodimer1 (P/N L3224 Thermo Fisher Scientific). In accordance with 10X standard procedure, a single-cell RNA library was generated using the GemCode Single-Cell Instrument (10x Genomics) and Single Cell 3' Library & Gel Bead Kit and Chip Kit (10x Genomics). The sequencing ready library was purified with SPRIselect, quality controlled for sized distribution and yield (LabChip GX Perkin Elmer), and quantified using qPCR (KAPA Biosystems Library Quantification Kit for Illumina platforms P/N KK4824). Finally, the sequencing was done using HiSeq2500 instrument (Illumina). Both the pre- and post-exercise libraries from subject #3 were re-sequenced to address the reproducibility of the analysis pipeline.

**Data pre-processing**. The Cell Ranger suite[43] (v3.1) of analysis was used to demultiplex sequencing results generated by Illumina sequencer into FASTQ files and subsequent alignment to reference genome, GRCh38. All following analytical steps were performed either in Python[44] (v3.7.0) using the Scanpy[45] v1.4.6 package or R[46] (v3.6.0) using Seurat[17] v3.0.1 package if not stated otherwise.

Data output from Cell Ranger was constituted by a total of 81,046 libraries. The Seurat pre-processing step of the scRNA-seq data included removal of low-quality cells by retaining the cells with ≥200 and ≤3000 expressed genes, and the exclusion of the genes expressed in ≤20 cells. To ensure that only high-quality cells are considered in the downstream analysis, cells with mitochondrial gene content ≥15% of the total genes detected were also excluded from further analysis. A high mitochondrial gene content is a feature associated with apoptotic and lysing cells. Thus, inclusion of cells with high mitochondrial gene-content could contaminate the pooled datasets with gene expressions linked to cell-death. The filtering steps excluded 12,035 cells (15%) from the original merged dataset, leaving 69,011 libraries. Global-scaling normalization method was used to account for differences in the library sizes by dividing gene-wise expression of each cell with the total expression, multiplication with the scaling factor (1e4), and log-transformation with a pseudo-count.

Considering gene cell to cell variability can greatly influence following downstream analysis, we have identified a subset of 2000 genes exhibiting high variability across the cells. Prior to performing an unsupervised dimensional reduction technique on the previously identified subset, the cell-gene matrix was Z-transformed to reduce the weight of the genes characterized by high expression. Based on inspection of the elbow-plot seven principal components (PCs) were considered as a representation of underlying data structure and used as input to batch balanced κ nearest neighbors (BBKNN v1.3.9) algorithm[47] to correct for confounding sources of variation. As the input dataset was large, BBKNN was used due to its benchmarked fast time performance at scale. The default parameters of BBKNN were used, as provided by Scanpy v1.4.6. Leiden graph clustering was performed on the BKNN neighborhoods and visualized with the UMAP algorithm[48,49].

Among the main drawbacks of suspension single cell sequencing methods, is the propensity to form doublets and multiplets. Traditionally, the impact of this has been addressed by library size filtering. To account for these technical artefacts and to improve the identification of DE genes, we implemented a doublet detection tool (R package DoubletFinder[50] v2.0.3). The doublet identification pipeline was applied to each sample individually following the pre-processing steps conducted with Seurat, using the no-ground-truth alternative of the pipeline. The doublet

formation rate used in the pipeline was assumed to be 7.5% and number of generated artificial doublets was set to default. Optimal neighborhood size parameter was selected using mean-variance normalized bimodality coefficient maximization. In total, 5175 doublets and 489 singlets belonging to the four clusters with a high degree of doublets infiltration (>60%) were identified and removed from further analysis.

**Differential expression**. Differential gene expression analysis was used to identify potential cell type specific markers and to assess the shift in gene expression of individual clusters as an effect of exercise using Wilcoxon signed-rank test. A gene was considered as a potential marker/distinguishing feature of a cluster, if it was significantly upregulated in the cluster, compared with the rest cell population ($fdr < 0.05$), and being detected in ≥50% of the cells in the cluster. Differential expression of the exercise effect was assessed by comparing cells within each cluster isolated before and after the exercise bout. A gene was deemed as differentially expressed at $fdr < 0.05$ and when expressed in >50% of the cells within the cluster, both before and after exercise bout.

**Annotation and ontologies**. Transcriptome-based annotations do not have a gold standard approach for definitively annotating a cluster. Therefore, the Leiden-based clusters generated above were annotated by a multi-step process, to strengthen the validity of the final annotations. Three main steps were included in the annotation scheme: first utilizing a cluster-based annotation toolkit, scCATCH[18] v2.0; second compiling cluster-specific marker genes (log2FC > 0, $fdr < 0.05$ and percentage of expressed cells ≥50%) which were compared against the CellMatch cell-type specific marker gene database; and third comparing the expression profiles of each cluster against cell-type specific transcriptome profiles provided by the Human Gene Atlas. Annotations which were consistent in ≥2 steps were used to give the final annotation of the cell clusters.

ScCatch is an annotation tool that parses transcriptome matrices and generates a ranked list of annotations. Following Leiden-based clustering, the gene matrices were parsed through scCatch. The top generated annotations were taken into consideration in the following steps of the annotation scheme. Cluster-specific marker genes (log2FC > 0, $fdr < 0.05$, and percentage of expressed cells ≥50%), which had been generated by differential gene expression between the clusters against all other clusters at baseline, were compared against the cell-type specific markers available in the CellMatch reference database[18]. Resultantly, a list of annotations was produced and compared against the scCatch results, to ensure that the annotations were consistent. Furthermore, the expression profiles of each cluster were also compared with cell-type specific transcriptome profiles provided by the Human Gene Atlas to strengthen the annotation validity. Potential increases in cellular abundance following exercise were assessed for each cell type by one-sided Wilcoxon tests.

To facilitate the biological interpretation of the exercise effect on each annotated cell type, gene set enrichment analysis was also performed using the R package ClusterProfiler[51] v3.10.1, adjusting for the background of the genes detected in each individual cluster. Solely terms with $fdr < 0.05$ were considered as significant for each given cluster. Thus, several biological terms were coupled with each cluster to strengthen the interpretability of the cell-type annotation.

**Principal curve analysis and exercise effect in muscle cells**. To characterize the transition from undifferentiated to more differentiated cells along myogenic lineage as an effect of acute exercise, a principal curve analysis was performed. From the dataset, three myogenic clusters were isolated and PCA was reperformed to update the shared PCA-space of the corresponding clusters. The first seven PCs were used to calculate two lineages, originating in the PAX7+ cluster and ending in the TNNI1+ and the TNNI2+ clusters, respectively. The R package princurve[52,53] v2.1.3 was used to calculate the non-parametric principal curves. A pseudotime value was defined as the coordinate when orthogonally projecting a cell onto the principal curve, as previously described by the slingshot pipeline[54]. In order to compare the transition in gene expression along each principal curve, the linear relationship between PC1 loading, containing the largest degree of variance, and normalized gene expression level were calculated for canonical myogenic cell markers. To quantify the overall shift along the pseudotime between pre- vs. post-exercise, empirical distribution functions (ECDFs) were calculated and compared using Wilcoxon signed-rank test.

**Comparison with whole-tissue RNA-sequencing data**. Comprehensive profile of gene expression in human skeletal muscle was defined using RNA-seq data obtained from GTEx database (release v7) consisting of 564 individuals. Library normalization to logCPM was done using edgeR package (v3.24.3). A total of 5844 genes with mean logCPM >4 were considered as detected in human skeletal muscle. For the single-cell analysis genes expressed in ≥50% cells and in at least one of the identified cell types with the mean log1p expression >0 were considered as detected in human skeletal muscle on the single-cell level, resulting with the total of 1946 unique genes. RNA-seq data pre and 3 h after a single bout of exercise was obtained from the recent publication by Norrbom et al.[22]. Genes differentially expressed at $fdr < 0.05$ were deemed to be exercise-regulated in the RNA-seq analysis.

**Skeletal muscle biopsy, myoblast isolation, and cell culture**. Myoblasts were isolated from a 25-year male as previously described[55] and stored in cryotank. Myoblasts were thawed and cultivated to expand and proliferate on Geltrex (A1413301, Thermo Scientific) coated plates in a serum-free chemically defined medium (P-CDM), modifications from WO 2010/031190 A1. P-CDM consist of three base media RMPI1640 (#11875093, Thermo Scientific), Ham's F12 (#11765054, Thermo Scientific) and MCDB120 (MBS652968, MyBioSource, Inc.) mixed 1:1:1, and supplanted with 2 mM L-Glutamine (#25030081, Thermo Scientific), 1X ITS-A (#51300044, Thermo Scientific), 1:2000 Fatty acid supplement (F7050, MERCK), 0.39 mg/ml Dexamethasone (D4902, MERCK), 0.5 mg/mL bovine serum albumin (A2153, MERCK), 0.5 mg/ml Fetuin (F3385, MERCK), 4 ng/ml bFGF (NBP2-34921, Novus Biologicals), 4 ng/ml FGF4 (NBP2-34864, Novus Biologicals), 4 ng/ml EGF (NBP2-34952, Novus Biologicals) and 4 ng/ml IGF-1 (#100-11, PeproTech). The myoblasts were replated on Geltrex-coated wells, and when cells reached 50% the myoblasts were harvested. For differentiation towards myotubes, the myoblasts were grown in a differentiation medium (D-CDM) which have the same base chemically defined medium as above but was supplemented with 2mM L-Glutamine (#25030081, Thermo Scientific), 1X ITS-A (#51300044, Thermo Scientific), 1:2000 Fatty acid supplement (F7050, MERCK), 0.5 mg/mL bovine serum albumin (A2153, MERCK), 0.5 mg/mL Fetuin (F3385, MERCK) and 4 ng/ml IGF-1 (#100-11, PeproTech) for four and nine days before they were harvested. All the harvested cells were washed with PBS, detached with TrypLE (#12605028, Thermo Scientific), transferred to tubes, and spun at 350 G for five minutes, supernatant removed, TRIzol reagent (#15596018, Thermo Scientific) added, and stored at −20 °C for further processing. Total RNA was isolated using the Direct-zol RNA Miniprep kit (R2052, Zymo Research) according to manufacturer protocol. The total RNA was measured in a DS-11 FX machine (DeNovix), and 60 ng of RNA was converted into cDNA using iScript™ Advanced cDNA Synthesis Kit (#1725038, Bio-Rad). Five nanograms of cDNA were used per SYBR green qPCR reaction with iTaq Univer SYBR Green Supermix (#1725124, Bio-Rad) using primers targeting the specific genes of interest in a CFX96 Touch Real-Time PCR Detection System (Bio-Rad). Expression levels were expressed as $2^{-(deltaCT)}$ arbitrary units in relation to GAPDH and differential expression was analyzed using one-way ANOVA with Tukey's honest significance test as post hoc test.

**C2C12-experiments**. Mouse immortalized myoblasts from the C2C12 cell-line exposed to proliferating ($n = 3$) and differentiating ($n = 3$) media analyzed on Affymetrix Mouse Expression 430 A Array were obtained through gene-expression omnibus (GDS2151) and analyzed for log2FC differential expression using LIMMA[56] where $fdr < 0.05$ was considered significant.

## Data availability

Raw- and processed sequencing data is publicly available through Gene-Expression omnibus (GEO) accession number GSE214544. Complete composition can be found in Supplementary Data 1. Complete marker-gene differential expression can be found in Supplementary Data 2, 3, and 4. Complete exercise-differential expression and ontology analysis can be found in Supplementary Data 5–9. Gene expression data from cell validation experiments can be found in Supplementary Data 10. The source data underlying Figs. 1d, 2d, 3a, 3d–h, 4b, c, 5a, b, e, f it can be found in Supplementary Data 10. Complete microarray gene-expression data from C2C12 cells can be found through GEO accession number GDS2151.

## Code availability

All code used in this paper has been deposited on GitHub and is available under https://github.com/HypoChloremic/scMuscle (https://doi.org/10.5281/zenodo.7126070).

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

## Acknowledgements

The authors acknowledge the Eukaryotic Single Cell Genomics (ESCG) facility in Stockholm funded by Science for Life Laboratory, KI Core, and StratRegen.

## Author contributions

E.R., T.G., R.F.G., M.A., A.L., and M.M. conceived the study and designed the experiments. E.R., T.G., R.F.G., M.A., A.L., and A.R. wrote the manuscript. E.R. supervised all data analysis and M.A. supervised in vitro validation experiment. M.M. scheduled human subject and supervised the exercise intervention, E.R. performed human muscle biopsies, S.A. preserved human muscle biopsies and S.A. and A.S. implemented purification technique. S.A. and R.F.G. performed in vitro validation experiment. A.L. and A.R. performed all data analysis for the manuscript. All authors have read and approved the final manuscript.

## Funding

## Competing interests

The authors declare no competing interests.
