## [Peer Review File · Communications Biology]

Reviewers' comments:

Reviewer #1 (Remarks to the Author):

1. Brief summary of the manuscript: This study employed single-cell RNA-seq technique to deconstruct human skeletal muscle before and after exercise in order to study muscle microenvironments with effects of exercise. The authors classified human skeletal muscle cells into 6 different cell types: endothelial cells, pericytes, mesenchymal cells, satellite cells, monocyte cells, and lymphocyte cells. The authors have pointed out some transcriptional changes after exercise.

2. Overall impression of the work: This is the first study using single-cell RNA-seq in human skeletal muscle in the context of exercise. This provides a resource for future study of human skeletal muscle to study effects of exercise. In terms of analysis to classify cell types, cell classification method with its rationale needs more clarification and justification to strengthen their analysis, as cell classification is one of major components of the study and interpretation of data. The validation experiments which are not included in the current study will significantly strengthen the conclusion of the study.

3. Specific comments, with recommendations for addressing each comment

3.1. Method sections is missing.

3.2. Line 50: Sentence is not complete: "... favorable response to ."

3.3. Line 117: Cell types were classified into six different types. There is a brief description of how these cells have been classified. However, it is very critical how the cell types were classified for further analysis and interpretation of the study. I recommend the authors to provide more detailed information with figure(s) about how these cell types were classified, by providing detailed information about marker genes with their statistics (i.e. expression level, p-value, fold change, adjusted p-value, % of cells in a cluster expressing a marker, etc.). Figures like Figures 2C and 2D to show marker expression throughout clusters are also fine. Also, I recommend the authors to show justification of the cell classification to address: Does there any other cell type exist beyond the suggested 6 cell types? (i.e. fibroblast?)

3.4. Figure 1A: Colors in UMAP are hard to distinguish between each sample based on current figure. I recommend to display UMAP of individual samples all separately, instead of all combined UMAP in a single UMAP.

3.5. Figure 1D: Unlike the authors' comments about reproducibility, Subject 3 and Subject 3 re-analysis do not appear to be very similar to each other, although those two should be expected to be highly similar. It is recommended to justify these difference in cell composition between these samples. Also, Subject 1 and Subject 3 are similar to each other, but others seem to be highly variable between each other. It is recommended to clarify this. In addition, does each subject include all samples (i.e. pre-exer., post-exer)? Is the difference of the cell composition between samples because of combining all samples for each subject?

3.6. Figure 1E: It needs to be more clearly explained about how "transcripts were detected" is defined across single-cell RNA-seq. Does it mean average number of detected genes across single cells in single-cell RNA-seq? Otherwise, does it mean the total number of genes detected across all single-cell RNA-seq? Does "detected" mean expression level greater than 0?

3.7. Figure 1C and Figure 2A: Authors need to clarify the difference between the two different UMAP.

3.8. Figure 2A: As the subclusters are classified based on a few markers particularly expressed in each subcluster, it is recommended to show each marker expression for subclusters (i.e. S100A8, S100A9, TNNI1, ...) using figures like Figure 2C and 2D, to clearly show that these markers are highly expressed in each subcluster.

3.9. Figure 2: It is recommended to term TNNI1+ satellite cells and TNNI2+ satellite cells as myogenic cells or mature muscle cells, etc., because satellite cells imply undifferentiated or proliferating progenitor cell populations.

3.10. Figure 2C-2D: For each cell type, one or two markers were shown, but it is recommended to show expression of multiple cell type markers at least 3 markers to show that the cell type classification is reliable. All markers expression does not need to be in main figures, and they can be

included in a supplementary figure.

3.11. Figure 3A: Bar graph showing each cell type in pre- and post-exercise is good. However, it would be clearer if the authors add each UMAP of pre- and post-exercise to visually show cell composition change between pre- and post-exercise.

3.12. Figure 3A: The authors have single-cell RNA-seq data for three subjects. Accordingly, cell type composition change can be compared with some statistics (i.e. p-value) and also can show which cell type composition is statistically significant (i.e. p-value < 0.05).

3.13. Line 216: A total of 874 genes were differentially expressed. : This number is sum of the number of differentially expressed genes from all 6 cell types. Some genes may be commonly differentially expressed across multiple cell types. Then, total number of differentially expressed genes should be lower than 874. The text needs to be clarified.

3.14. Line 208 and 598: Hours post exercise are not consistent. In line 208, three hours after exercise, but 4 hours after exercise in line 598.

3.15. It is recommended to include data to validate single-cell RNA-seq results. For example, the authors may quantify cell type composition on tissue biopsies to validate effects of exercise.

Reviewer #2 (Remarks to the Author):

This work uses scRNAseq to describe how physical exercise changes the composition of muscle tissue cells and how it affects their transcriptomic profile. The authors profiled muscle biopsies before and after physical exercise from 3 subjects and constructed a single-cell atlas to perform differential gene expression and trajectory analysis. In particular they describe 3 different populations of muscle satellite cells and gene markers associated to them. While I think there is great value in generating and thoroughly describing such single-cell datasets, many technical questions concerning the analysis limit my appreciation of the findings.

1. I just wonder if n=3 subjects is enough, given one of them is significantly older than the other 2. The authors don't seem to describe if they see any age-dependent effects. I think it could also be interesting to include the effect of age and maybe have 3 'young' vs 3 'older' subjects. This would greatly strengthen the paper.

2. Figure 1D: maybe show the total number of cells per subject and separate pre and post exercise since that seem to be one of the take-home messages of the paper.

3. The choice of integration method is important and can lead to very different outcomes/grouping of cells. I recommend the authors to briefly explain why they picked ref 17 over other methods (Harmony, Scanorama, etc). I would also add technical details about integration in the methods section were the authors already do a good job at explaining their data processing steps.

4. I'm a bit sceptical about the annotation using the CellMatch database. In my experience scCatch will always find a cell type even without a precise match in the database. Consequently, I'm not convinced the three populations of satellite cells are all indeed satellite cells. Only one of the three express PAX7 (Fig 2C). The authors also don't detect myonuclei from mature muscle fibers, which depending on the sample prep method should be captured. Are the TNNT1 and TNNT2 positive populations truly satellite cells? Or mature myonuclei? I would try to map the expression of myosin light chain MYL1 to confirm that and perhaps include it in the genes of Figure 2D. I think this is a very important point to clarify.

5. The authors also used the Human Gene Atlas to label the clusters. They should show which marker genes they picked like for Figure 2D.

6. In supplementary, I would include a table of all differentially expressed gene from the clusters they

resolve, prior to any annotation.

7. Figure 2A: The skeletal muscle cell type subclusters UMAP is informative. However, what is the difference between Fig 2A (left UMAP) and Fig 1C?

8. Figure 2B: What does the circle radius represent?

9. Figure 2C: What is the expression range? Scale bar? If it's binary, what the threshold?

10. A UMAP of pre and post exercise could be nice to complement Fig 3A because it can also illustrate a shift in transcriptomic profile.

11. Figure 3D: Including a boxplot in the violin plot might help the reader appreciate the statistical differences between pre and post. What is expression metric on the vertical axis? (same for Fig. 4D and E).

12. I would focus on satellite cells and show a volcano plot of differentially expressed genes. In supplementary I would also be curious to see a table of differentially expressed genes pre vs post of each of 6 cell types.

13. Line 301: What does higher level of differentiation mean in this context?

14. Figure 4D/E: What does the horizontal axis represent? Pseudotime? Why negative values?

Minor/other points:

- Single-cell vs single cell (consistency)
- Line 99: Maybe the table can indicate the number of single-cell transcriptomes recovered per subject
- Some typos throughout the manuscript
- Line 108: was the muscle or the library sequenced twice? Subject 3 had 4 biopsies taken?
- Line 289: 2nd or 3rd PC? Fig 4A shows 3rd.

Reviewer #1 (Remarks to the Author):

Brief summary of the manuscript: This study employed single-cell RNA-seq technique to deconstruct human skeletal muscle before and after exercise in order to study muscle microenvironments with effects of exercise. The authors classified human skeletal muscle cells into 6 different cell types: endothelial cells, pericytes, mesenchymal cells, satellite cells, monocyte cells, and lymphocyte cells. The authors have pointed out some transcriptional changes after exercise.

Overall impression of the work: This is the first study using single-cell RNA-seq in human skeletal muscle in the context of exercise. This provides a resource for future study of human skeletal muscle to study effects of exercise. In terms of analysis to classify cell types, cell classification method with its rationale needs more clarification and justification to strengthen their analysis, as cell classification is one of major components of the study and interpretation of data. The validation experiments which are not included in the current study will significantly strengthen the conclusion of the study.

Thank you for your kind words and suggestions.

Specific comments, with recommendations for addressing each comment

3.1. Method sections is missing.

Thank you for pointing this out, the methods section was submitted as a supplement due to word-count restrictions from the journal. If the reviewers and editor think it would be beneficial, we are more than happy to move it into the actual manuscript.

3.2. Line 50: Sentence is not complete: "... favorable response to."

Thank you.

3.3. Line 117: Cell types were classified into six different types. There is a brief description of how these cells have been classified. However, it is very critical how the cell types were classified for further analysis and interpretation of the study. I recommend the authors to provide more detailed information with figure(s) about how these cell types were classified, by providing detailed information about marker genes with their statistics (i.e. expression level, p-value, fold change, adjusted p-value, % of cells in a cluster expressing a marker, etc.). Figures like Figures 2C and 2D to show marker expression throughout clusters are also fine. Also, I recommend the authors to show justification of the cell classification to address: Does there any other cell type exist beyond the suggested 6 cell types? (i.e. fibroblast?)

Thank you. We have extended the methods section to clearly state and explain our considerations in the cell-type annotation. In our mind, we have chosen a quite conservative annotation strategy with a combination of scCatch, cell-atlas, and curated marker genes. The rationale is that given the novelty of this methodology in the context of skeletal muscle physiology we prioritize validity over resolution i.e., we allow adjacent subpopulations of cells to be annotated into a broader defined cell-type rather than putting forward

numerous uncertain subpopulations than over time risk turning out to be invalid. Therefore, populations of fibroblasts are contained in the mesenchymal cluster and subpopulations such as fibro-adipogenic progenitors (as put forward by Rubenstein et al). In our study no subpopulation of mesenchymal cells was unambiguously annotated as fibroblasts, FAP or other subpopulation and we therefore choose to label them as mesenchymal cells.

Furthermore, we have now extended the supplementary information. In Supplementary tables 5-8, all marker genes, in addition to all other features detected are presented in detail for the cell types and subpopulations (including expression level, p-value, fold change, adjusted p-value, % of cells in the cluster expressing the marker).

3.4. Figure 1A: Colors in UMAP are hard to distinguish between each sample based on current figure. I recommend to display UMAP of individual samples all separately, instead of all combined UMAP in a single UMAP.

Thank you for the suggestion and we agree that given the high variance in cell-retrieval the contribution of each sample is hard to distinguish on the UMAP presented in the manuscript, but we also think that putting 6 individual UMAPs in a figure takes a lot of space without adding much information to the reader. To this end we have now produced a down-sampled UMAP showing a proportional down-sampled version of cells (randomly selected) from each sample. In our mind, this gives a good representation of the relative contribution of each sample to each cluster with good readability. Individual crude, (not down-sampled) UMAPS from each subject are presented as Supplementary figure 1. It is important to point out that even though the cell-retrieval varied considerably from sample-to-sample (which is always the case when cells are isolated from solid fibrous tissue), the cellular composition (i.e., % cells in each population) was consistent.

3.5. Figure 1D: Unlike the authors' comments about reproducibility, Subject 3 and Subject 3 re-analysis do not appear to be very similar to each other, although those two should be expected to be highly similar. It is recommended to justify these difference in cell composition between these samples. Also, Subject 1 and Subject 3 are similar to each other, but others seem to be highly variable between each other. It is recommended to clarify this. In addition, does each subject include all samples (i.e. pre-exer., post-exer)? Is the difference of the cell composition between samples because of combining all samples for each subject?

Thank you for your comments. In Figure 1D, the baseline cellular compositions are presented, i.e., pre-exercise bout for each subject. This has now been clarified in the figure legend. Exercise-effects are shown in Figure 3-5.

We thank the reviewer for pointing out the lack of consistency between the re-sequenced sample #3 in Figure 1 and this was due to a mistake on our side. The reason for this discrepancy was that Subject 3 and subject 3 re-sequenced were not matched with regards to the retrieved UMI tag per cell. The same filtering requirements were used when analyzing all subjects, irrespective of read-depth. Thus, cells with different UMIs were filtered out, when comparing subject 3 and subject 3 re-sequenced, due to differences in read-depth. Consequently, the composition of subject 3 and subject 3 re-sequenced were different, due

to the use of the same filtering requirements. In the updated figure 1D, the UMIs were matched between subject 3 and subject 3 re-sequenced. The resulting cellular composition show that the annotations are a perfect match.

3.6. Figure 1E: It needs to be more clearly explained about how “transcripts were detected” is defined across single-cell RNA-seq. Does it mean average number of detected genes across single cells in single-cell RNA-seq? Otherwise, does it mean the total number of genes detected across all single-cell RNA-seq? Does “detected” mean expression level greater than 0?

Thank you for your comment. We agree with the reviewer that this section requires more detailed clarification. We have now updated materials and methods section:

Whole-tissue RNA-sequencing data from human skeletal muscle tissue was obtained from version 7 of the GTEx database and pre-processed using edgeR package. Genes with the mean logCPM values >4 were considered as detected in human skeletal muscle resulting with total number of 5844 unique transcripts. For single-cell sequencing, genes found to be expressed in ≥50% of cells in at least one of the identified cell types with the mean log₁₀ expression >0 were considered as detected in human skeletal muscle on the single-cell level, resulting with the total of 1946 unique genes.

3.7. Figure 1C and Figure 2A: Authors need to clarify the difference between the two different UMAP.

We agree it was not optimal to have two versions of the UMAPs in the manuscript and we have now harmonized the coordinate-calculations of the UMAPs so that they are consistent.

3.8. Figure 2A: As the subclusters are classified based on a few markers particularly expressed in each subcluster, it is recommended to show each marker expression for subclusters (i.e. S100A8, S100A9, TNNI1, ...) using figures like Figure 2C and 2D, to clearly show that these markers are highly expressed in each subcluster.

We have now added a Supplementary figure 2, where all mentioned marker genes are presented, both regarding expression across the UMAP and comparing all subclusters with violin plots for the normalized expression level. This further elucidates the differential expression of the marker genes for the given subpopulations.

3.9. Figure 2: It is recommended to term TNNI1+ satellite cells and TNNI2+ satellite cells as myogenic cells or mature muscle cells, etc., because satellite cells imply undifferentiated or proliferating progenitor cell populations.

Thank you for the suggestion, we agree that satellite cells can be a bit misleading in this context and has changed the naming to myoblasts which we hope is a clearer, yet fairly well-defined denomination and we have changed it accordingly.

3.10. Figure 2C-2D: For each cell type, one or two markers were shown, but it is recommended to show expression of multiple cell type markers at least 3 markers to show

that the cell type classification is reliable. All markers expression does not need to be in main figures, and they can be included in a supplementary figure.

Yes, we agree, but we also think it would be too spacious to add more figure-panels to the paper. In the previous version one could actually find all markers genes for all clusters in the supplementary (quite extensive) tables, but we appreciate that this does not suffice and have therefore added this information to the Supplemental table 5.

3.11. Figure 3A: Bar graph showing each cell type in pre- and post-exercise is good. However, it would be clearer if the authors add each UMAP of pre- and post-exercise to visually show cell composition change between pre- and post-exercise.

Thank you for the suggestion. We have added such UMAPs as Supplementary figure 1 however, given the very small (which is to be expected given it is only 3 hours between the samples) compositional changes we do not think it is of enough interest to be put in a main figure panel.

3.12. Figure 3A: The authors have single-cell RNA-seq data for three subjects. Accordingly, cell type composition change can be compared with some statistics (i.e. p-value) and also can show which cell type composition is statistically significant (i.e. p-value < 0.05).

Thanks for the suggestion, the significant increase of circulating cells in the post-exercise biopsy has now been added.

3.13. Line 216: A total of 874 genes were differentially expressed. This number is sum of the number of differentially expressed genes from all 6 cell types. Some genes may be commonly differentially expressed across multiple cell types. Then, total number of differentially expressed genes should be lower than 874. The text needs to be clarified.

Thank you for the suggestion. Indeed, total number of 874 differentially expressed genes reflects the sum of upregulated genes across six cell types. This is now adjusted in the manuscript for the shared genes across the cell types. We have also included Supplementary tables 5-9 of upregulated genes together with corresponding fold change.

3.14. Line 208 and 598: Hours post exercise are not consistent. In line 208, three hours after exercise, but 4 hours after exercise in line 598.

This was a typo on our side. Thank you.

3.15. It is recommended to include data to validate single-cell RNA-seq results. For example, the authors may quantify cell type composition on tissue biopsies to validate effects of exercise.

We deem that a general staining-based quantification of the different cell-types is not feasible in human muscle samples given the low density of most of the cell-types (and even more so for subpopulations). For instance, satellite cells only constitute 1-2% of the cells in a

cross section, meaning an average of <1 cell per section. We completely agree with the reviewer that the myoblast subpopulations should be addressed further. To this end we have now conducted an additional experiment where myoblasts have been isolated from human muscle biopsies and the mRNA-levels of the key marker genes that was identified in the single-cell sequencing been analyzed over several stages of differentiation using RT-PCR. In addition, we re-analyzed a publicly available microarray experiment of mouse myoblasts undergoing differentiation which allowed us to compare gene-expression between immature proliferating myoblasts with differentiating myoblasts. The results have been added to the paper. In summary, we show that both human and rodent-derived myoblasts initiate an ~30-fold increase in mRNA expression of TNNC1 and TNNC2 when put into differentiation media. This serves as an in vitro validation of the findings from the single-cell sequencing by showing that the key markers genes of the subpopulations identified as differentiating are indeed regulated as a part of the differentiation process. The cell-isolation protocol used herein with straining and lysis-steps was specifically chosen in order to isolate mononucleated cells, not myonuclei.

Reviewer #2 (Remarks to the Author):

This work uses scRNA-seq to describe how physical exercise changes the composition of muscle tissue cells and how it affects their transcriptomic profile. The authors profiled muscle biopsies before and after physical exercise from 3 subjects and constructed a single-cell atlas to perform differential gene expression and trajectory analysis. In particular they describe 3 different populations of muscle satellite cells and gene markers associated to them. While I think there is great value in generating and thoroughly describing such single-cell datasets, many technical questions concerning the analysis limit my appreciation of the findings.

1. I just wonder if n=3 subjects is enough, given one of them is significantly older than the other 2. The authors don't seem to describe if they see any age-dependent effects. I think it could also be interesting to include the effect of age and maybe have 3 'young' vs 3 'older' subjects. This would greatly strengthen the paper.

We agree with the reviewer that muscle ageing processes and sarcopenia are important topics, but we also think that it is not feasible to cover both healthy muscle baseline characterization, acute exercise responses in healthy subjects, and, on top of that, analyzing ageing-related processes. This is the first study to address exercise effects across different cell-types in healthy muscle and we hope it will suffice to be recognized as a contribution and hope we will be able to conduct future studies specifically designed to address effects of disease and ageing utilizing the techniques employed in the present study. We agree three subjects is a small sample and it is of course not sufficient to address inter-individual differences, but it is sufficient to detect the traits that are common and thus represents an internally valid transcriptional response to a single bout of exercise. While this is the first single-cell sequencing based study on exercise, there are hundreds of publications describing the transcriptional response to exercise on whole muscle. Based on the variance observed on the whole-tissue RNA-sequencing data used for validation in the present study, we estimate that an average exercise effect of 1.8 (fold-change) can be detected at alpha 0.05 with n=3. Even though one might expect a larger degree of technical noise in single-cell sequencing, this method also constitutes an extensive oversampling, since a large number of cells from each cell population are sequenced, wherefore one should be able to detect a large number of exercise-regulated genes with n=3. The rationale for including healthy, physically active subjects of both sexes and with a certain age-span was to increase external validity, i.e., the effects we see should apply to a healthy adult population. To our knowledge no study has ever reported age-dependent differences in acute exercise responses when comparing 25-30 year-olds with healthy 50 year-olds. The (relatively few) studies showing age-differences compare elderly subjects (>65 years) with young adults, and in those studies the transcripts reported to be differently regulated in the elderly constitute a very small portion of the overall transcriptional response to exercise. Thus, even though there are some differences between elderly subjects and young adults in the transcriptional response to exercise, the overall, global transcriptional response is highly reproducible. This can be seen also in meta-analysis².

Nevertheless, we agree that our study has important limitations in this regard and have added a paragraph highlighting this in relation to sample size and potential effects of age and sex.

2. Figure 1D: maybe show the total number of cells per subject and separate pre and post exercise since that seem to be one of the take-home messages of the paper.

Thank you for the suggestion, we have now updated Supplementary table 1.

3. The choice of integration method is important and can lead to very different outcomes/grouping of cells. I recommend the authors to briefly explain why they picked ref 17 over other methods (Harmony, Scanorama, etc). I would also add technical details about integration in the methods section were the authors already do a good job a explain their data processing steps.

Thank you for your comments and suggestions. When choosing the data processing pipeline, there were several aspects considered. The Scanpy and Seurat packages were used for data processing. To ensure a reproducible pipeline, the anchoring methods had to therefore be compatible to each package, resultantly subsetting the packages that were befitting of use. Furthermore, as the experiments resulted with >60 000 cells, with sparse gene matrices including ~17 000 genes, the data processing pipeline needed to be well performing in terms of processing speeds. To evaluate the available packages, these were finally tested against the datasets generated in this manuscript. BBKNN alignment method, which is a well-cited and suggested for optimal performance by the Scanpy authors, was highly performant both in terms of iterative speed and processing time when run against the data presented in this manuscript. One additional aspect of consideration was the open-source availability of the package code and the possibility to further iterate and raise issues when those arose during the analysis process. Considering Scanpy presented a well-integrated solution with BBKNN, it further motivated it as a choice of anchoring method.

4. I'm a bit sceptical about the annotation using the CellMatch database. In my experience scCatch will always find a cell type even without a precise match in the database. Consequently, I'm not convinced the three populations of satellite cells are all indeed satellite cells. Only one of the three express PAX7 (Fig 2C). The authors also don't detect myonuclei from mature muscle fibers, which depending on the sample prep method should be captured. Are the TNNT1 and TNNT2 positive populations truly satellite cells? Or mature myonuclei? I would try to map the expression of myosin light chain MYL1 to confirm that and perhaps include it in the genes of Figure 2D. I think this is a very important point to clarify.

We agree that methodological considerations taken, in particular with regards to cell-type annotation, needed clarification in the manuscript. This has now been added and is quite extensively described. We believe we have taken the risk of misclassification by scCatch (and we agree with the reviewer that scCatch can be overly optimistic in this regard) into account and have utilized a quite conservative strategy with regards to cell type classification with a combination of scCatch, cell-atlas and curated marker genes where we marked a cluster as unambiguously classified only if at least two of these methods were in agreement. The rationale is that given the novelty of this methodology in the context of skeletal muscle physiology we prioritize validity over resolution i.e., we allow adjacent subpopulations of cells to be annotated into a broader defined cell-type rather than putting forward numerous uncertain subpopulations than over time risk turning out to be invalid.

On that note, we also agree that the identified subpopulations of myoblasts of varying degree of differentiation should be validated. To this end we have now conducted an additional experiment where myoblasts have been isolated from human muscle biopsies and the mRNA-levels of key marker genes were analyzed over several stages of differentiation using RT-PCR. In addition, we re-analyzed a publicly available microarray experiment of mouse myoblasts undergoing differentiation. In summary, we show that both human and rodent-derived myoblasts initiate mRNA expression of TNNT1 and TNNT2 when put into differentiation media. This serves as an in vitro validation of the findings from the single-cell sequencing by showing that the key markers genes of the subpopulations identified as differentiating are indeed regulated as a part of the differentiation process. The cell-isolation protocol used herein with straining and lysis-steps was specifically chosen in order to isolate mononucleated cells, not myonuclei, were lysed cells and isolated nuclei are spun out. All samples were checked microscopically for cell viability prior to analysis and no remaining debris or isolated nuclei were detected.

5. The authors also used the Human Gene Atlas to label the clusters. They should show which marker genes they picked like for Figure 2D.

Thank you for your suggestion. We have now added a Supplementary Figure 2, in addition to Supplementary tables 5-8, where the marker gene expressions are presented in detail (e.g., in terms of expression level across the UMAP visual landscape, in comparison to each subpopulation, and the statistics regarding the differential expression).

6. In supplementary, I would include a table of all differentially expressed gene from the clusters they resolve, prior to any annotation.

We have now added Supplementary tables 5-8, where all features are presented in detail (e.g., in regard to fold change, p-value, expression level, comparing the subpopulations and against each other and within each subpopulation pre- and post-exercise).

7. Figure 2A: The skeletal muscle cell type subclusters UMAP is informative. However, what is the difference between Fig 2A (left UMAP) and Fig 1C?

This was due to differences in Scanpy and Seurat-based UMAP-coordinates. We have now harmonized the UMAP-coordinates throughout the manuscript.

8. Figure 2B: What does the circle radius represent?

The circle radius represents the q-value of the given term. This has now been explained in the figure legend.

9. Figure 2C: What is the expression range? Scale bar? If it's binary, what the threshold?

Thank you for your comment. We have now added the normalized expression level cutoff used for coloring the plot. The cutoffs can be compared against the adjacent violin plots, further elucidating where the marker gene expression appears.

10. A UMAP of pre and post exercise could be nice to complement Fig 3A because it can also illustrate a shift in transcriptomic profile.

Thank you for this suggestion. We have now added Supplementary figure 1 however, given the very small shifts in composition we do not think it is reasonable to put them as a main figure.

11. Figure 3D: Including a boxplot in the violin plot might help the reader appreciate the statistical differences between pre and post. What is expression metric on the vertical axis? (same for Fig. 4D and E).

Thank you for this suggestion. The expression metric on the vertical axis is the normalized gene expression, and we have now further explained this in the figure legends. We believe that the violin plot gives a higher resolution visualization of the normalized gene expression distribution for the given genes and cell types.

12. I would focus on satellite cells and show a volcano plot of differentially expressed genes. In supplementary I would also be curious to see a table of differentially expressed genes pre vs post of each of 6 cell types.

Figure 5D is a scatterplot (which is essentially a volcano but with discrete coding of significance rather than continuous) of differentially regulated genes after exercise in the three myogenic subpopulations. A detailed differential expression tabulation for all cell-types are also included in the supplementary tables 5-8.

13. Line 301: What does higher level of differentiation mean in this context?

Thank you for this comment. We have now clarified this statement to “The continuous transition of cells towards higher expression of genes associated with differentiation” to become more distinct in data versus the interpretation.

14. Figure 4D/E: What does the horizontal axis represent? Pseudotime? Why negative values?

The pseudo-time calculation is based on principal components so the axis are PC1 (x-axis) and normalized expression level (y-axis) and by default the PCA is scaled with positive and negative numbers. We agree with the reviewer that from a biological standpoint it makes more sense to use positive numbers and have now changed the x-axis to reflect this.

Minor/other points:

- *Single-cell vs single cell (consistency)*

Thank you for your comment. This was now adjusted.

- *Line 99: Maybe the table can indicate the number of single-cell transcriptomes recovered per subject*

Thank you for your suggestion. We have taken this under consideration and provided information in Supplementary table 1, rather than as part of Table 1.

- *Some typos throughout the manuscript*

Thank you for your comment. This was now corrected.

- *Line 108: was the muscle or the library sequenced twice? Subject 3 had 4 biopsies taken?*

Thank you for your comment. We agree this section requires better clarification. We have now corrected the manuscript.

- *Line 289: 2nd or 3rd PC? Fig 4A shows 3rd.*

Thank you for your comment. This was now corrected.

References

1. Deane, C. S. *et al.* The acute transcriptional response to resistance exercise: Impact of age and contraction mode. *Aging (Albany, NY)*. **11**, (2019).
2. Su, J. *et al.* A novel atlas of gene expression in human skeletal muscle reveals molecular changes associated with aging. *Skelet. Muscle* **5**, (2015).

Reviewers' comments:

Reviewer #1 (Remarks to the Author):

Thank you for addressing reviewer concerns. In this revision, the authors adequately clarified their materials and methods section as well as main texts that I asked for clarification in the review at the authors' first submission. The revised figures are now more understandable. More detailed descriptions on the authors' methods helped to understand the authors' rationales clearly. Hence, I believe that this revision adequately addressed all concerns that I have previously pointed out.

Reviewer #2 (Remarks to the Author):

The authors have clarified most of my technical questions about the data analysis and I am satisfied about their changes. Thank you very much for the detailed response.

My remaining concerns are about cell annotation and the trajectory model.

I think some clarifications are needed concerning the three myogenic subpopulations you resolve. First of all, myoblasts and muscle satellite cells are regarded as different cell types. The introduction (third paragraph) state that there are basically the same. Could this be nuanced? Quiescent SC are typically Pax7+ cells, and upon exercise or injury, they activate myogenic programs and asymmetrically divide into MyoD+ myogenic progenitor or myoblast. Myoblasts then expand in numbers, mature into Myog+ myocytes, and ultimately fuse to form new myofibers. This review details the cellular players of the myogenic lineage and their markers: <https://doi.org/10.1007/s00018-019-03093-6>

Further, this single-cell paper, although in mouse, also provides a model of the myogenic lineage built from scRNAseq muscle regeneration data and presents different types of cells and their markers: <https://doi.org/10.1016/j.celrep.2020.02.067>

I would check the markers they use to define SCs, myoblasts, and more mature myogenic progenitors, and perhaps see where they map on your three myogenic clusters. Injury and exercise activate similar myogenic programs.

I would therefore not group the three myogenic clusters together in Fig 2D. I would maybe include Myog, Myod1, and Acta1 (mature myonuclei) as well in the violin plots. I understand you are trying to merge some similar cluster together, and you have a nice supplementary figure with all the individual clusters. But in the case of the myogenic clusters I'm afraid you are mixing very distinct populations together. At the very least keep the Pax7+ distinct from the other two. From your supplementary tables, these genes seem differentially expressed across your three myogenic subpop. Perhaps the Pax7+ cluster is a mix of SCs and early myoblasts (Myf5+), while the two TNN+ cluster terminally differentiated muscle fibers (probably myonuclei) and not myoblasts. You also mention MYF5 in your trajectory analysis which is an important marker of early activated SC. I think it is important to clarify the nomenclature in accordance with what is considered the current ground truth in the muscle stem cell field.

I would probably call the PAX7+ cluster (Activated SC and myoblasts).

Concerning the trajectory analysis, I was wondering if you had tried other inference models such as Slingshot or Monocle? I don't think there is anything wrong with your method, although perhaps a bit too simplistic. From what I understand, you seem to calculate the principal curve only from 2 principal components. Other methods infer trajectories and assign pseudo-time values to single cells from higher dimensional data and may be more accurate. If the authors have time, I would strongly encourage them to try these established methods.

Importantly as well, how is the pseudo-time defined? (I suppose you don't have pseudo time values

per single cell). Could you please include how you define the pseudo-time axis in your methods section? Thank you very much.

Could you also please label the axes of Fig 5 A and B.

The authors have clarified most of my technical questions about the data analysis and I am satisfied about their changes. Thank you very much for the detailed response.

My remaining concerns are about cell annotation and the trajectory model.

I think some clarifications are needed concerning the three myogenic subpopulations you resolve. First of all, myoblasts and muscle satellite cells are regarded as different cell types. The introduction (third paragraph) state that there are basically the same. Could this be nuanced? Quiescent SC are typically Pax7+ cells, and upon exercise or injury, they activate myogenic programs and asymmetrically divide into MyoD+ myogenic progenitor or myoblast. Myoblasts then expand in numbers, mature into Myog+ myocytes, and ultimately fuse to form new myofibers. This review details the cellular players of the myogenic lineage and their markers: <https://doi.org/10.1007/s00018-019-03093-6>

Further, this single-cell paper, although in mouse, also provides a model of the myogenic lineage built from scRNAseq muscle regeneration data and presents different types of cells and their markers: <https://doi.org/10.1016/j.celrep.2020.02.067>

I would check the markers they use to define SCs, myoblasts, and more mature myogenic progenitors, and perhaps see where they map on your three myogenic clusters. Injury and exercise activate similar myogenic programs.

I would therefore not group the three myogenic clusters together in Fig 2D. I would maybe include Myog, Myod1, and Acta1 (mature myonuclei) as well in the violin plots. I understand you are trying to merge some similar cluster together, and you have a nice supplementary figure with all the individual clusters. But in the case of the myogenic clusters I'm afraid you are mixing very distinct populations together. At the very least keep the Pax7+ distinct from the other two. From your supplementary tables, these genes seem differentially expressed across your three myogenic subpop. Perhaps the Pax7+ cluster is a mix of SCs and early myoblasts (Myf5+), while the two TNN+ cluster terminally differentiated muscle fibers (probably myonuclei) and not myoblasts. You also mention MYF5 in your trajectory analysis which is an important marker of early activated SC. I think it is important to clarify the nomenclature in accordance with what is considered the current ground truth in the muscle stem cell field.

I would probably call the PAX7+ cluster (Activated SC and myoblasts).

We would like to thank the reviewer for the initiated and encouraging feedback and suggestions during the review-process. We agree that the different muscle populations found are indeed distinct cell-types rather than the more method-specific and contemporary term sub-populations, even though one can discuss the semantics of the difference between 'cell-type' and 'subpopulation'. The rationale for using the same overall nomenclature for the three muscle cell populations in the first part of the paper was merely to keep the number of distinct labels down as much as possible when discussing the more global findings on what cell-types was identified. The differences and possible trajectories between the three clusters are then addressed in more detail later in the manuscripts (figure panel 4 and 5). Regardless, we have now separated the muscle clusters into the PAX+ and TNN+ clusters and label them as per the reviewers suggestion. We cannot exclude that the TNN+ clusters are, at least partly, formed from myonuclei rather than late

stage/differentiating myogenic cells but it is clear from both in vitro data presented herein and datasets from differentiating C2C12 cells that TNN-expression (also ACTA1 and DES) occur at earlier stages of differentiation than fusion into fibers or tubes. The cell-debris removal method used to isolate the mononucleated cells is specifically designed (and has, at least for heart tissue been shown to work <https://www.miltenyibiotec.com/US-en/products/debris-removal-solution.html>) to deplete the sample of muscle fiber organelles including nuclei. During the microscopic evaluation for viability and cell-count prior to library-prep on the 10X no isolated nuclei was noted. But again, we understand the reviewers concern and have added appropriate references and a paragraph to the discussion highlighting the uncertainty of whether the TNN+ clusters are late-stage differentiating myocytes (PAX7- low MYOD and low Myogenin-expression) or terminally differentiated myonuclei, or a mixture of both.

Concerning the trajectory analysis, I was wondering if you had tried other inference models such as Slingshot or Monocle? I don't think there is anything wrong with your method, although perhaps a bit too simplistic. From what I understand, you seem to calculate the principal curve only from 2 principal components. Other methods infer trajectories and assign pseudo-time values to single cells from higher dimensional data and may be more accurate. If the authors have time, I would strongly encourage them to try these established methods. Importantly as well, how is the pseudo-time defined? (I suppose you don't have pseudo time values per single cell). Could you please include how you define the pseudo-time axis in your methods section? Thank you very much.

Thank you for your comment. The trajectory analysis used is very much inspired by slingshot but to a larger extent hardcoded, to a large degree for our own learning-purposes and to improve on our understanding of the methodology. For the dimensionality reduction we used seven PCs, this has been amended to the methods section (even though as the reviewer seems well aware of, the exact number of dimensions has very little impact on the result). We performed a validation of the trajectory analysis using the slingshot library, (we have attached the result below), which produced identical trajectories. Also, inference of pseudotime for each observation was also carried out the same way as in slingshot (princurve-based using orthogonal projections) which has been clarified in the methods-section. Thank you for your constructive feedback!

Could you also please label the axes of Fig 5 A and B.
Of course, sorry for the inconvenience.

REVIEWERS' COMMENTS:

Reviewer #2 (Remarks to the Author):

Thank you for addressing my points. I really enjoyed reviewing your work and I have not additional comments to make.